

**Intra- and inter-annual changes in isoprene emission from central Amazonia**
Eliane Gomes Alves[1,2*], Raoni Aquino Santana[3], Cléo Quaresma Dias-Junior[4,2], Santiago
Botía[5], Tyeen Taylor[6], Ana Maria Yáñez-Serrano[7,8,9], Jürgen Kesselmeier[10], Pedro Ivo
Lembo Silveira de Assis[11], Giordane Martins[11], Rodrigo de Souza[12], Sergio Duvoisin
Junior[13], Alex Guenther[14], Dasa Gu[15], Anywhere Tsokankunku[16], Matthias Sörgel[16], Bruce
Nelson[17], Davieliton Pinto[11], Shujiro Komiya[1], Diogo Martins Rosa[11], Bettina Weber[18,10],
Cybelli Barbosa[10,18], Michelle Robin[1], Kenneth J Feeley[19], Alvaro Duque[20], Viviana
Londoño Lemos[21], Maria Paula Contreras[22], Alvaro Idarraga[23], Norberto López A.[23], Chad
Husby[24], Brett Jestrow[24].
[1] Department of Biogeochemical Processes, Max Planck Institute for Biogeochemistry, Jena, Germany
[2] Climate and Environment Department, National Institute of Amazonian Research, Manaus, Brazil
[3] Department of Atmospheric Sciences, Federal University of Western Para, Santarem, Brazil
[4] Federal Institute of Para, Belem, Brazil
[5] Department of Biogeochemical Signals, Max Planck Institute for Biogeochemistry, Jena, Germany
[6] Department of Civil & Environmental Engineering, University of Michigan, USA.
[7] 7IDAEA-CSIC, 08034, Barcelona, Spain
[8] CREAF, E08193 Bellaterra (Cerdanyola del Vallès), Catalonia, Spain
[9] Global Ecology Unit, CREAF-CSIC-UAB, E08193 Bellaterra (Cerdanyola del Vallès), Catalonia, Spain
[10] Multiphase Chemistry Department, Max Planck Institute for Chemistry, Mainz, Germany
[11] Department of Tropical Forest Sciences, National Institute for Amazonian Research, Manaus, Brazil
[12] Meteorology Department, State University of Amazonas, Manaus, Brazil
[13] Chemistry Department, State University of Amazonas, Manaus, Brazil
[14] Department of Earth System Science, University of California, Irvine, U.S.A.
[15] Division of Environment and Sustainability, Hong Kong University of Science and Technology, Clear
Water Bay, Hong Kong, China
[16] Atmospheric Chemistry Department, Max Planck Institute for Chemistry, Mainz, Germany
[17] Coordination of Environmental Dynamics, National Institute of Amazonian Research, Manaus, Brazil
[18] Institute for Biology, Division of Plant Sciences, University of Graz, Graz, Austria
[19] Department of Biological Sciences, University of Miami, Coral Gables, FL, USA.
[20] Departamento de Ciencias Forestales, Universidad Nacional de Colombia–Sede 19Medellín, Medellín,
Colombia
[21] Department of Plant and Microbial Biology, University of Minnesota, USA
[22] Jardín Botánico de Cartagena "Guillermo Piñeres", Turbaco, Bolívar, Colombia.
[23] Fundación Jardín Botánico de Medellín, Antioquia, Colombia.
[24] Fairchild Tropical Botanic Garden, Miami, FL, USA
*egomes@bgc-jena.mpg.de











**Abstract**
Isoprene emissions are a key component in biosphere-atmosphere interactions, and the
most significant global source is the Amazon rainforest. However, intra- and inter-annual
variations in biological and environmental factors that regulate isoprene emission from
Amazonia are not well understood and, thereby, poorly represented in models. Here, with
datasets covering several years of measurements at the Amazon Tall Tower Observatory
(ATTO), in central Amazonia, Brazil, we (1) quantified canopy profiles of isoprene mixing
ratios across seasons of normal and anomalous years and related them to the main drivers
of isoprene emission – solar radiation, temperature, and leaf phenology; (2) evaluated the
effect of leaf age on the magnitude of the isoprene emission factor ($E_s$) from different tree
species and scaled up to canopy with intra- and inter-annual leaf age distribution derived
by a phenocam; and (3) adapted the leaf age algorithm from MEGAN with observed
changes in $E_s$ across leaf ages. Our results showed that the variability in isoprene mixing
ratios was higher between seasons (max. during the dry-to-wet transition seasons) than
between years, with values from the extreme 2015 El-niño year not significantly higher
than in normal years. In addition, model runs considering in-situ observations of canopy $E_s$
and the modification on the leaf age algorithm with leaf-level observations of $E_s$ presented
considerable improvements in the simulated isoprene flux. This shows that MEGAN
estimates of isoprene emission can be improved when biological processes are
mechanistically incorporated into the model.
**1. Introduction**
Isoprene dominates the emission of biogenic volatile organic compounds (BVOCs) into
the atmosphere, and its major global source is tropical vegetation (Guenther et al., 2012;
Sindelarova et al., 2014). In the atmosphere, isoprene is a short-lived (minutes to hours)
reactive BVOC species, and its photooxidation affects the atmospheric oxidation capacity
contributing to the formation of ozone ($O_3$) and secondary organic aerosols (SOA)
(Atkinson, 1997; Pöschl et al., 2010). With its high plant foliage biomass and rich plant
diversity (ter Steege et al., 2013), the Amazon Forest represents a key source of isoprene
to the atmosphere (Yáñez-Serrano et al., 2020). However, model estimates of isoprene
emission and its intra- and inter-annual variability in the Amazon still carry high
uncertainty, because only a few observational experiments have been conducted with
mechanistic and process-based approaches, which hinders further modeling optimization
(Alves et al., 2018; Yáñez-Serrano et al., 2020). One of the most critical knowledge gaps
is how plants' isoprene emission differs under extremely hot and dry conditions, such as in
El-niño years, and how this might affect atmospheric processes. As some studies have
indicated that extreme years will become more frequent and intense with climate change
(Nobre et al., 2016; Boulton et al., 2022), it is essential to understand the processes
mediated by isoprene in such years to improve model estimates (Yáñez-Serrano et al.,
2020; Artaxo et al., 2022).
Some reasons for uncertainties in isoprene model estimates are already known. The correct
determination of the magnitude of the isoprene source - or the emission factor at leaf
standard conditions (1000 $\mu$mol m$^{-2}$ s$^{-1}$ photosynthetically active radiation- PAR, 30 °C),
as it is conceptualized in models (e.g., Guenther et al., 1995) - is crucial to improve isoprene



modeling estimates. The Amazon plant biodiversity represents a considerable challenge for determining the isoprene emission factor. Although previous studies suggested that ~ 1% of tree species are hyperdominant - with their tree individuals responsible for half of all tree stems, carbon storage, and productivity (ter Steege et al., 2013; Fauset et al., 2015) -, it is still unclear which plant species can emit substantial amounts of isoprene (Monson et al., 2013), how these isoprene emitters are distributed throughout the Amazon basin, and how the isoprene emission factor varies seasonally and interannually as result of changes in eco-physiological processes (Gomes Alves et al., 2022). Another source of uncertainty is related to quantifying the main sinks of isoprene. Once emitted by plant foliage, isoprene can undergo surface deposition onto plant canopy (Karl et al., 2004) and soil (Pegoraro et al., 2006), can be oxidized at rates depending on the atmospheric concentration of other gases such as $NO_x$, $O_3$ and OH (Atkinson, 1997), and can be transported into and out of the atmospheric boundary layer (Wei et al., 2018). Additionally, the rapid conversion of isoprene photooxidation products can open a further sink for BVOCs in plants. This chemical and biological processing of emitted compounds may affect vertical transport processes, again influencing the biosphere (Kesselmeier et al., 2002; Canaval et al., 2020).

In addition, seasonal variation in isoprene emission from Amazon forests has been reported by several in-situ studies, with the indication that isoprene seasonality is driven by intra-annual variation in solar radiation, temperature, and leaf phenology (Kuhn et al., 2004a, b; Yáñez-Serrano et al., 2015; Alves et al., 2016, 2018; Wei et al., 2018; Langford et al., 2022). On a larger scale, satellite retrievals of isoprene oxidation products, like formaldehyde (Barkley et al., 2009; Bauwens et al., 2016), and direct retrieval of isoprene (Fu et al., 2019; Wells et al., 2022) have given an initial view of the long-term Amazon isoprene emission, enabling not only seasonal but also inter-annual comparisons. Yet, there remains a need to parameterize and evaluate the estimations with local and regional measurements and to gain a better understanding of the main processes related to sources and sinks of isoprene, since some studies have shown that satellite-derived isoprene emission values are either overestimated (Alves et al., 2016) or underestimated (Gu et al., 2017), or even show maximum emissions in a different season when compared to in-situ measurements (Alves et al., 2016, 2018).

Here we report in-situ observations of isoprene mixing ratios during different seasons and in consecutive years in central Amazonia to evaluate intra- and inter-annual variabilities in two normal years (2013-2014) and one El-niño year (2015); in addition, we report observations of leaf-level isoprene emission factor and leaf phenology monitoring. With the intra- and inter-annual observations of isoprene at a central Amazonian site, this study proposes to: (1) quantify the isoprene mixing ratios across seasons of normal and anomalous years and compare them with the main drivers of isoprene emission – solar radiation, temperature, and leaf phenology; (2) evaluate the effect of leaf age on the magnitude of the isoprene emission factor from different tree species and scale up with canopy intra- and inter-annual leaf age distribution; and (3) use the Model of Emissions of Gases and Aerosols from Nature (MEGAN) to assess the effects of the observed changes in the isoprene emission factor across leaf ages, by modifying the leaf age algorithm and comparing simulations with observations at canopy-level.





## 2. Methods

*2.1 Amazon Tall Tower Observatory (ATTO)*

We performed measurements at the ATTO site located 150 km northeast of Manaus in the Uatumã Sustainable Development Reserve (USDR) in central Amazonia. The climate is tropical humid, with two distinctive seasons – wet season (December-May) and dry season (July-October) and transition seasons in between – and has a mean annual precipitation of 2380 mm (Botía et al., 2022). The vegetation in this area is considered mature, mostly non-flooded rainforest (terra-firme), with a mean canopy height of 35 m, and predominantly occurs on plateaus at a maximum altitude of approximately 130 m a.s.l. (Andreae et al., 2015). Utmost air masses arriving at the site come from the east (NE~20%, ENE~27%, E~33%, ESE~19%) (Zannoni et al., 2020) and have passed through 1500 km of undisturbed terra-firme rainforest, with minor intrusion of air masses from Manaus (Pöhlker et al., 2019). Figure 1 shows seasonal variation in solar radiation, air temperature, precipitation, and soil moisture from 2013 to 2019. Andreae et al. (2015) have more details on this experimental site.

*2.2 Mixing ratios of isoprene – canopy level*

Isoprene gradient mixing ratios were inferred by air samples collected from the INSTANT tower (80 m height, coordinates: S 02°08.7520' W 58°59.9920') at eight heights in and above the canopy (0.05, 0.5, 4, 12, 24, 38, 53 and 79 m) during intensive campaigns across different seasons from November 2012 to October 2015. Eight heated (50 ºC) and insulated inlets (fluorinated ethylene propylene - FEP) were connected to a Proton Transfer Reaction – Mass Spectrometer (PTRMS) (Ionicon Analytic GmbH, Austria), which was housed in an air-conditioned container 10 m from the INSTANT tower. The inlets were guided to a valve system, switching every 2 min between the different heights, completing a full profile in 16 min. The mean total uncertainty of isoprene mixing ratios was 9.9 %, within the PTRMS measurement uncertainty (~10%). For more details on the experimental setup, PTRMS conditions, and calibration, we refer the reader to Yãnez-Serrano et al. (2015).

*2.3 Flux of isoprene – canopy level*

During a campaign in November 2015, eddy covariance fluxes of isoprene were measured for 11 days. Isoprene concentrations were obtained with the above-described PTRMS at a time resolution of 1 s and from a separate 3/8" inlet at 41 m height that sampled air at a flow rate of about 10 l min$^{-1}$. A CSAT3 sonic anemometer (Campbell Scientific Inc., Logan, U.S.A.) measured the three-dimensional wind speed at high frequency and was placed at a distance of 0.5 m from the isoprene inlet. Fluxes were then calculated by correlating fluctuations of the vertical wind vector to the fluctuations of isoprene concentrations with the software package EddyPro® (LI-COR Inc., Lincoln, U.S.A.). A method for despiking and raw data statistical screening was employed (Vickers and Mahrt, 1997). Half-hourly averaged fluxes were flagged according to a method of data quality control (Mauder and Foken, 2004), and only data with the highest quality (flags 0 and 1) was used for further analyses. Losses for sampling frequencies between 0.1 and 0.8Hz have



been observed as below 10% (Guenther and Hills, 1998; Spirig et al., 2005; Holst et al.,
2010; Jensen et al., 2018). Footprints were calculated using a two-dimensional model for
a geographic domain of 2 x 2 km centered at the INSTANT tower (Kljun et al., 2015). The
Tovi Footprint Analysis Toolbox (LI-COR Inc., Lincoln, U.S.A.) was used to calculate
half-hourly footprints and to combine them for the measurement period. More details on
the flux measurements and data processing are given in Pfannerstill et al. (2018).
*2.4 Leaf Area Density – measurements with the Light Detection and Ranging sensor*
*(LiDAR)*
Measurements of canopy leaf area density were carried out with a ground Light Detection
and Ranging sensor (LiDAR) at the ATTO site. These measurements aimed to give
information on the canopy structure around the INSTANT tower. Ground-LiDAR surveys
were conducted in October 2015 with a Riegl LD90-3100VHS-FLP system (Horn,
Austria), which generated a canopy profile map in vertical and horizontal directions. We
walked ten transects of 150 m in length with the ground-LIDAR system, and measurements
were averaged every 15 m of each transect, summing up to ten measurements per transect.
Measurements of all ten transects were then averaged and presented with the confidence
interval (95%). More details about how the ground LiDAR data were analyzed can be
obtained from Stark et al. (2012).
*2.5 Leaf-level monitoring of leaf demography and phenology*
Leaf demography and phenology of 36 trees were monitored from March 2016 to
December 2017. Along 100 m of canopy walkways, canopy leaves were monitored
monthly to determine leaf ages and investigate how leaf age proportions vary during the
year. Ten branches of each tree were randomly selected and labeled with one iron ring at
their bottom end. All leaves attached from the bottom to the apical end were counted and
dated according to the day of observation. For the first observation, all leaves were assigned
with unknown age. In the following months, every time a new leaf was observed, the date
of observation was recorded for that specific leaf. For leaf age determination, the date of
the first observation of a new leaf was set back to 15 days before observation. The age was
calculated by the difference, in the number of days, between the first day and the last day
of observation, resulting in a number of days with a deviation of plus-minus 15 days. For
instance, if a new leaf was observed on 1st July 2017, the flushing date of this leaf was
assigned for 17th June 2017 (+/- 15 days). Then, all subsequent measurements considered
17th June 2017 as a date for leaf flushing, and aging was counted based on the number of
days that this leaf stayed attached to the branch.
*2.6 Isoprene emission factor – leaf level*
Leaves of 21 canopy tree species, out of the 36 trees monitored for leaf demography and
phenology (described in section 2.5), were measured to determine the isoprene emission
factor across different leaf ages (Table S1) from October to November 2017. The other 15
trees were unreachable with the sampling system and, therefore, not measured. Leaf-level
isoprene sampling was carried out in 2-3 leaves of each age class available for each tree





during the measurement period, using a commercial portable gas exchange system GFS-
3000 (Walz, Effelthich, Germany). Each leaf was separately enclosed in the leaf chamber
at standard conditions – photosynthetic photon flux density (PPFD) set to 1000 μmol m$^{-2}$
s$^{-1}$ and leaf temperature to 30ºC - until net assimilation, stomatal conductance and internal
$CO_2$ concentration were stable. The measurement stability criterion was assigned as one
standard deviation of the net assimilation mean. The airflow rate going into the leaf
chamber was 400 μmol s$^{-1}$, and $CO_2$ and $H_2O$ concentrations were 400 μmol.mol$^{-1}$ and 21
mmol.mol$^{-1}$ (relative humidity of ~60%), respectively. Air exiting the GFS-3000 leaf
chamber was routed to fill sorbent cartridges (stainless steel tubes filled with Tenax TA
and Carbograph 5 TD sorbents), and a downstream pump sampled the exiting air at a rate
of 200 sccm for 10 min. A hydrocarbon filter (Restek Pure Chromatography, Restek
Corporations, USA) was installed at the air inlet of GFS-3000 to remove isoprene from the
incoming ambient air, and all tubing in contact with the sampling air was made of PTFE.
Before each measurement, a blank sample was obtained from the empty leaf chamber.

Isoprene content in the sorbent cartridges was determined by laboratory analysis at the
University of California (Irvine, U.S.A.). All cartridges were placed into a thermally
desorbing autosampler (TD-100, Markes International, Inc). The isoprene was pre-
concentrated at 10 °C followed by injection into a gas chromatograph (GC, model 7890B,
Agilent Technologies, Inc) equipped with a time-of-flight mass spectrometer (Markes
BenchTOF-SeV) and a flame ionization detector (TD-GC–FID/TOF-MS) (Woolfenden
and McClenny, 1999; ASTM D6196-15, 2015). Internal standards tetramethylethylene
and decahydronaphtalene were injected into each sample after collection and before
analysis. The system was calibrated daily with a commercial isoprene standard from Apel
Riemer Environmental Inc. The external gas standard was prepared using a dynamic
dilution system, and the effluent was added to sorbent cartridges under conditions similar
to those used for sampling. Once the volume mixing ratio of isoprene (ppbv) was obtained,
leaf emission flux was determined using the Eq. (1):

$$F = R_{ppbv} \times \frac{Q}{A} \qquad (1)$$


where $F$ (nmol.m$^{-2}$.s$^{-1}$) is leaf flux of isoprene emission; $R_{ppbv}$ (nmol.mol$^{-1}$) is isoprene
concentration of the sample (cartridge); $Q$ is the flow rate of air into the leaf chamber (400
μmol.s$^{-1}$); and $A$ is the area of leaf within the chamber (0.08 m²). The isoprene emission
rate was then calculated and converted to mg.m$^{-2}$.h$^{-1}$. For more details on tree species, leaf
age, and assigned leaf age class, see Table S1 in Supplementary Information.

*2.7 Tower-camera derived leaf phenology and demography data*

Upper canopy leaf phenology was monitored with a Stardot RGB camera (model Netcam
XL 3MP) installed at 81m height on the ATTO INSTANT tower. For more details on the
camera setup, radiometric calibration, and detection of phenological stages, we refer the
reader to Lopes et al. (2016). Only images acquired near noon and under an overcast sky
(diffuse illumination) were selected for subsequent analysis. The camera (subsequently
called phenocam) monitored the upper crown surfaces of 194 trees from July 2013 to



November 2018. Images were analyzed to track the temporal trajectory of each tree crown
and assign them into one of three classes: "leaf flushing" (crowns that showed a strong
increase in greening), "leaf abscising" (crowns which showed a large increase in greying,
which is the color of bare upper canopy branches) or "no change". By counting the number
of individual trees per month for each category (flushing or abscission), we aggregated our
census to the monthly scale. Of the monitored trees, 69% (n = 134) had clear flushing and
abscission patterns, and, using the number of days after each flushing event, we determined
leaf age classes and attributed a fraction of the upper canopy crowns to an age class at
monthly intervals. We defined the following leaf age classes: (i) young leaves (0−1
month), (ii) growing (1−2 months), (iii) mature leaves (3−6 months), and (iv) old leaves
(>6 months). Then, we partitioned the age classes into classes of leaf area index (LAI)
(i.e., young, growing, mature, and old LAI) by normalizing each leaf age class with the
total LAI measured at ATTO. A constant LAI of 5.32 $m^2$ $m^{-2}$ was used for all months,
since the variability of this number throughout the year was not statistically significant
(unpublished results). For the normalization, we considered the total number of trees in
the camera frame (n = 194), assuming that the 31% that do not have clear flushing patterns
are part of the old age class. For more details on the methods and assumptions for
separating LAI into leaf age classes, see Wu et al. (2016). Datasets of flushing and
abscission (http://doi.org/10.17871/ atto.223.7.840) and the raw LAI age classes (http://doi.
org/10.17871/ atto.230.4.842).
*2.8 Isoprene emission trait – tree species level*
To get more detailed information on the trees monitored with the camera, a total of 194
trees were taxonomically identified, and the isoprene emission trait was assigned. Isoprene
emission data were obtained from published data and new measurements for the study
species. New measurements were conducted at the ATTO research site (described in
section 2.6), and additional measurements were obtained using the PORCO method (Taylor
et al., 2021), a customized photoionization detection system, on trees in tropical botanical
gardens. Briefly, all PORCO measurements were made in situ on uncut 'sun' branches by
enclosing one-to-few leaves inside rigid leaf cuvettes, acclimating them to darkness, and
then exposing the leaves to photosynthetically active radiation controlled at 1000 μmol $m^{-2}$
$s^{-1}$, and temperatures near 30˚C, for 3.5 minutes of measurement time. Emission rates
were corrected to a 30˚C equivalent based on a standard temperature response curve
(Guenther et al., 1993). Emission rates exceeding 1 nmol $m^{-2}$ $s^{-1}$ were considered positively
indicative of isoprene emissions. See the full method validation and a discussion of the
rarity of detection of other compounds as false positives for isoprene in Taylor et al. (2021).
Botanic gardens used for tree measurements were: A. Duque private collection, Retiro,
Antioquia, Colombia; Fairchild Tropical Botanical Garden, Miami, FL, USA; Jardín
Botánico de Cartagena "Guillermo Piñeres", Turbaco, Bolívar, Colombia; Jardín Botánico
"Joaquín Antonio Uribe" de Medellín, Antioquia, Colombia; Montgomery Botanical
Garden, Miami, FL, USA; Universidad Nacional de Medellín–Sede Medellín arboretum,
Antioquia, Colombia.
For applying isoprene measurements from external datasets (botanic garden measurements
or published literature) to our study species, we followed the methods of Taylor et al.,



(2018, 2019). We used data compiled from 12 literature sources (Bracho-Nunez et al.,
2013; Geron et al., 2002; Harley et al., 2004; Keller & Lerdau, 1999; Klinger et al., 1998;
Klinger et al., 2002; Lerdau & Keller 1997; Padhy & Varshney, 2005; Tambunan et al.,
2006; Taylor et al., 2018; Taylor et al., 2021; Varshney & Singh, 2003). Tree species
taxonomy was standardized by the Taxonomic Name Resolution Service (Boyle et al.,
2013; Boyle et al., 2021). We assigned species data only in terms of the genetically
determined capacity to produce isoprene (Monson et al., 2013); we did not consider the
variability in the strength of emissions, for which data are more limited and potentially
confounded by method variation and species plasticity.A species-level emission status–
emitter or non-emitter–was applied where available in external datasets; otherwise, genus-
level information was used to impute the emission status to unmeasured species. The
proportion of measured species in a genus that emit isoprene was used as an estimate of
the probability ($p$IE) that any species sampled from the genus would be an emitter.  For a
genus corresponding to one of our study species, for $p$IE $\leq 1/3$, the species was estimated
to be a non-emitter, and for pIE $\geq 2/3$, the species was estimated to be an emitter. For values
$1/3 < p$IE $> 2/3$, the genus average was considered ambiguous and the species was excluded
from the analyses. Whereas there is some expected error in the assignment of emission
status to any given species, analyses of large numbers of species will tend toward the
correct answer due to the tendency of genera to predominate in emitting or non-emitting
species (Taylor et al., 2018). All species for which no emission data were available at the
genus level were excluded from the analyses. The imputed isoprene emission status and
associated information for each of our study species can be found in Table S2. The source
data (literature reference or present study metadata) for each species that informed the
imputation can be found in Table S3.

*2.9 Modeled isoprene flux estimates - Model of Emissions of Gases and Aerosols from*
*Nature (MEGAN)*

Isoprene fluxes were simulated using the MEGAN version 2.1 model in which the flux
activity factor for isoprene ($\gamma$i) is proportional to the emission response to light ($\gamma$P),
temperature ($\gamma$T), leaf age ($\gamma$A), soil moisture ($\gamma$SM), leaf area index (LAI), and $CO_2$
inhibition ($\gamma CO_2$) according to Eq. (2) (Guenther et al., 2012):

$$\gamma i = C_{CE} LAI \gamma P \gamma T \gamma A \gamma SM \gamma CO_2 \qquad (2)$$

For this study, the canopy environment model of Guenther et al. (2006) was used with a
canopy environment coefficient ($C_{CE}$) of 0.57.  MEGAN was run accounting for variations
in light, temperature, and LAI fractionated into leaf age classes.  $CO_2$ inhibition and soil
moisture activity factors were set equal to a constant of 1, assuming these parameters do
not vary. For all simulations, we assumed no seasonal variation in soil moisture because
the soil moisture observed in this site consistently exceeds the threshold for the isoprene
drought response in MEGAN 2.1 (Guenther et al., 2012), which means that MEGAN would
predict no variation in isoprene emission resulting from the observed changes in soil
moisture (Fig. 1).



Solar radiation (PPFD) and air temperature inputs for all model simulations were obtained
from measurements at the INSTANT tower. Air temperature at 36 m height above ground
level was measured with a temperature and relative humidity sensor (CS215-L, Campbell Scientific
Inc., Logan, Utah, USA). In cases where the air temperature measurement at 36 m height failed,
the missing data were gap-filled with air temperature data available at other heights (73 m, 55 m,
40 m, 12 m), measured with CS215-L sensors installed on the INSTANT tower, or with the air
temperature at 18 m above the ground measured with a thermocouple (Conatex, St. Wendel,
Germany), installed along one evergreen tree of the species *Buchenavia parvifolia* (Combretaceae),
located 95 m away from the INSTANT tower. In cases where all the air temperature sensors failed
for less than 4 hours, the missing air temperature at 36 m height was gap-filled by linear
interpolation, visually checking data quality. In cases where no air temperature measurement was
available for a long time (e.g., one day, 2 months etc.), confirmed several times in 2013, the missing
air temperature at 36 m height was gap-filled by a multiple regression model developed with three
predictor variables: half-hourly variation of the soil temperature at 10 cm depth, soil heat flux, and
volumetric soil water content at 40 cm depth. The model training period was from 2013 June to
2014 May because the three predictor variables were usually available through the one-year period.
The developed model was validated based on the observation dataset from June 2014 to May 2015,
which showed good agreement with observed air temperature data at 36 m height during the
validation period ($R^2$ = 0.83; RMSE = 1.21; n = 7473). The developed and validated model was
applied to the three predictor variables measured in 2013 for gap-filling the long-term missing data
of air temperature at 36 m height. In cases where the predictor variables were unavailable in 2013,
the missing data were gap-filled using Akima interpolation with visual data quality checks.
Incoming and outgoing shortwave radiation was measured with a net radiometer (NR- Lite2, Kipp
& Zonen, the Netherlands) at 75 m above ground. In cases where the radiation measurement failed
for no more than 1 hour, the missing radiation data were gap-filled by linear interpolation, visually
checking data quality. In cases where radiation data were unavailable for more than 1 hour, the
missing data were gap-filled by the mean diurnal course (over ±15-day) method. Lastly, we used
a constant value (5.32) for the LAI and normalized it with monthly leaf age fractions
obtained from the phenocam observations to derive the canopy leaf age for each month
(see section 2.6). More details on model settings are found in Guenther et al. (2012).
**3. Results and Discussion**
*3.1 Observations of canopy isoprene mixing ratios*
We observed intra- and inter-annual variability of isoprene mixing ratios in canopy profiles
from nine intensive campaigns from Nov 2012 to Oct 2015 (Fig. 2a and Table 1). Figure
2b shows the leaf area density profile measured around the INSTANT tower in Oct 2015
and the mean canopy height. In general, isoprene mixing ratios were higher during the dry-
to-wet transition season (Nov 2012) and the dry season (Aug 2014 and Oct 2015/El-niño
year) than the wet season (Feb and Mar in 2013 and 2014) and the wet-to-dry transition
season (Jun 2013); with an exception for the Sep 2013-dry season that showed values
comparable to the 2014-wet season, although still higher than the 2013-wet season.
Interestingly, mean isoprene mixing ratios in Oct 2015 (El-niño dry season) were slightly
higher than those observed in Aug 2014 and Sep 2013 (both dry seasons) but not higher
than those observed in Nov 2012 (dry-to-wet transition) (although this was variable and
not significant). Seasonal changes in isoprene mixing ratios and fluxes from central
Amazonia have already been reported and were related to variations in temperature, light



availability at the surface, and leaf phenology (Yáñez-Serrano et al., 2015; Alves et al.,
2016, 2018; Wei et al., 2018; Langford et al., 2022), but the assessment of inter-annual
variability of consecutive years including anomalous years was lacking. Considering the
increased air temperatures observed in the 2015-El-niño dry season (Fig. 1b) and the fact
that tropical plant species emit high amounts of isoprene at high temperatures (Harley et
al., 2004; Alves et al., 2014; Jardine et al., 2014, Garcia et al., 2019; Rodrigues et al., 2020),
one could expect considerably higher emission and thereby high air mixing ratios of
isoprene during this extreme year. However, the 2015-El-niño dry season might have been
stressful for plants, with the anomalous drought (see soil moisture reduction in Fig. 1 d)
likely offsetting the high-temperature stimulus on isoprene emission. This finding can be
supported by two studies performed on this study site. Firstly, isoprene emission measured
in hyperdominant tree species showed a reduction in emission from the wet to the dry
season with a compensating increase in emission of monoterpenes and sesquiterpenes that
have both temperature-dependent emissions, indicating that the reduction in isoprene
emission and the shift toward heavier compounds resulted from abiotic stresses (e.g.,
drought) during the dry season (Gomes Alves et al., 2022), which might be substantially
higher in an extreme El-niño year. Secondly, the anomalous post-drought leaf flush
observed in Feb-Mar 2016 suggested that trees flushed out new leaves to recover from the
stress suffered during the 2015-El-niño dry season (Gonçalves et al., 2020).
Another interesting result was the seasonal variation in the shape of the isoprene mixing
ratio profiles (Fig. 2a). In general, all wet seasons (Feb-Mar 2013/2014) and the wet-to-dry
transition season (Jun 2013) data showed a constant profile with no clear vertical gradient
of isoprene. On the other hand, the dry seasons (Sep 2013, Aug 2014, and Oct 2015)
showed maximum mixing ratios between 12 m and 24m, and the dry-to-wet transition
season (Nov 2012) presented a well-defined peak at 24 m. This variation in the shape of
the isoprene mixing ratio profiles could result from a combination of variations in the
canopy leaf area density profile and canopy leaf age distribution throughout the year. The
total amount of LAI has a small variation over the year; still, the fractions of leaf ages that
compose this total LAI changes seasonally (Wu et al., 2016), as well as the shape of the
canopy leaf area density profile, with significant changes at the upper canopy (Martins
Rosa, 2016). During the wet-to-dry transition season (May-Jun) and the dry season (Jul-
Oct), upper canopy trees presented leaf abscission and leaf flushing (Lopes et al., 2016,
Gonçalves et al., 2020), and the maturing process on the following months toward the
beginning of the wet season (Nov-Jan) might translate into higher leaf area density at the
upper canopy (Martins Rosa, 2016) and higher gross primary productivity (GPP) fluxes
(Botía et al., 2022). This implies that two processes might be simultaneously occurring:
one is that when there are more leaves at the upper canopy, less light penetrates the canopy,
which might induce the maximum isoprene emission at the upper canopy as observed in
Nov 2012; the other one is that leaves at the upper canopy can have higher photosynthesis
rates and, consequently, a higher isoprene emission factor when they are mature (Alves et
al., 2014), and more mature leaves and higher GPP were observed in this study site during
the dry-to-wet transition season and beginning of the wet season (Lopes et al., 2016;
Gonçalves et al., 2020; Botía et al., 2022).



In addition, it has been suggested that seasonal variations in isoprene emissions could result
from a variation in the isoprene emission factor with leaf aging, but there were not enough
observational studies to support it in the Amazon (Alves et al., 2018). Therefore, in the
next section, we show for the first time in-situ observations of isoprene emission factor at
leaf-level with known leaf age and infer how this, together with variation in canopy leaf
age distribution, likely affected intra- and inter-annual variability in emission during
sequenced years.
*3.2 Seasonal changes in the isoprene emission factor ($E_s$)*
The isoprene emission factor ($E_s$; parameter measured at 1000 $\mu$mol m$^{-2}$ s$^{-1}$ PAR, 30 °C) of
an ecosystem is determined by the fraction of species that emits this compound and by
variations in the $E_s$ magnitude within species. Such variations may be conditioned by leaf
ontogenetic status (e.g., young leaves have no or low emission, and old leaves emit less
isoprene than mature leaves) and environment (e.g., sun-leaves have higher $E_s$ than shade-
leaves) (Niinemets, 2016). We performed measurements of $E_s$ from sun-adapted leaves
across different ages in 21 trees (from 20 tree species) located at the upper canopy and
around the tower, and values ranged from 0 to 3.52 mg m$^{-2}$ h$^{-1}$ (see all species and emission
values in table S1). Of these 21 trees, 60 % had isoprene emission detectable by our
analytical system (TD-GC-TOFMS), while the other 40% did not.  To evaluate whether
the $E_s$ changes with leaf aging, we calculated the $E_s$ ratios of mature (3−6 months) to young
(0−1 month), growing (1−2 months), and old (>6 months) leaves within the same tree
individual. We observed that, for some trees, $E_s$ can be reduced by half when leaves are
older than six months (Fig. 3 and table S1), but the average of all trees combined showed
a statistically significant $E_s$ reduction of 36% in old leaves compared to mature leaves
(paired t-test, p-value <0.05).
As tropical species represent a mix of phenotypes with the predominance of non-deciduous
plants, it was impossible to sample all leaf age classes for all tree species measured.
Nevertheless, our dataset covers leave ages from 15 to 578 days (table S1), and we observed
that all leaves measured at the young leaf age class did not show detectable isoprene
emission, and two leaves measured at the growing leaf age class showed emissions similar
to the mature leaf age class (Fig.3 and table S1). As our sampling did not cover a broad
range of leaf ages below 60 days, especially among isoprene emitters, to improve the
robustness of our analysis, we added another species that had the $E_s$ measured from the leaf
flushing day until the 30$^{th}$ day (young class) and at 226-227 days (old class) in the
southwestern Amazonia (Kuhn et al., 2004b). With this tree species added, we calculated
that the emission activity of $E_s$ of young (0−1 month) and old (>6 months) leaves were,
respectively, 1% and 64% of the $E_s$ observed in growing (1−2 months) and mature leaves
(3−6 months) (paired t-test, p-value <0.05), and that there was no statistically significant
difference between growing and mature leaves (paired t-test, p-value >0.05) (Fig. 3 and
table S1).
Furthermore, we observed that emitter species from our dataset could be combined into
two qualitative emission categories – medium emitter and low emitter –, given their $E_s$
magnitude compared to other leaf-level measurements in Amazonia (see a detailed



compilation in Yãñez-Serrano et al., 2020), and high emitter, with the data from the tree
species measured in southwestern Amazonia (Kuhn et al., 2004b) (Fig. 3). The maximum
$E_s$ occurred in different leaf ages for each emitter category. Still, both high and medium
emitters had an $E_s$ maximum before 150 days (mature). In contrast, the low emitter category
showed an $E_s$ maximum in 295 days (old) for one species, but that was not statistically
significant when compared to all low emitter species (paired t-test, p-value >0.05).
Therefore, this indicates that species that emit considerable amounts of isoprene have
maximum $E_s$ when their leaves are mature.
The variation of $E_s$ across leaf ages is already known, also for tropical tree species (Kuhn
et al., 2004b; Alves et al., 2014); however, the quantification of these variations across
different species is still a challenge given the high biodiversity in the Amazonian rainforest,
and, although our results show the variation of $E_s$ across leaf ages for more species than
previously reported, it is still necessary to further develop tools to upscale these results to
the ecosystem level. Earlier studies indicated that the capacity to emit isoprene is more
common, and the $E_s$ magnitudes are expected to be the highest in deciduous tree species
(Harrison et al., 2013; Dani et al., 2014). In fact, the high emitter (Fig. 3) is a tropical
deciduous tree species with a large range of variation in $E_s$ within 30 days after leaf flushing
and with the maximum $E_s$ observed in mature leaves at the end of the dry season (Kuhn et
al., 2004b). However, the number of deciduous trees that have regular leaf abscission and
leaf flushing during the dry season in central Amazonia may represent less than 15% of the
whole tree assembly (Gonçalves et al., 2020), which means that the effect of high
variability in the $E_s$ with leaf aging from those trees might be low at the ecosystem level,
especially when we compare it with the other trees that showed less variability in the $E_s$
(Fig. 3, table S1).
Furthermore, for Amazonian tree species, the categorization of phenological habits goes
beyond evergreen and deciduous. Here, with a dataset of 194 trees (Fig. 4, and table S2)
monitored with a phenocam for leaf phenology and demography from 2013 to 2018, we
derived: (i) the camera-based canopy leaf area index (LAI) fractionated into four leaf age
classes - young (<=1 month), growing (1-2 months), mature (3-6 months), and old (>6
months) (Fig. 4a); and (ii) four classes of phenology (phenotypes) - evergreen, semi-
evergreen, brevi-deciduous, and semi-brevideciduous (Fig. 4c), based on the frequency of
events of leaf abscission and leaf flushing (more details in Supplementary Information).
Then, we assigned the isoprene trait for these tree species with measurements and literature
data, and imputed the trait to non-measured species by following the method described in
Taylor et al. (2018) (Fig. 4 c). We observed that the isoprene trait did not have a higher
percentage within brevi-deciduous and semi-brevideciduous phenotypes, which have
regular and seasonal leaf abscission and leaf flushing. Instead, all phenotypes had a similar
fraction of isoprene emitters (Fig. 4c). This implies that leaf age is an important factor for
the magnitude of $E_s$ regardless of phenotype.
Although we do not have enough data to infer the phenotypes for the species monitored at
the branch level, we observed that the leaf age distribution of the 36 trees (Fig. 4b) was
similar to the 194 trees monitored with the phenocam (Fig. 4a); and that the fraction of
isoprene emitters was also similar when measured (21 trees – 60% emitters; Fig. 3) and
non-measured (15 trees – 47% emitters) were combined (56% emitters) (Fig. 4d) and





compared to the phenocam trees (60% emitters) (Fig. 4c). Note that the tree species that
had no isoprene emission trait reported in the literature and did not fill the assumptions
necessary to input the trait, according to Taylor et al. (2018), were assigned with the
unknown flag (NA).
The similarity found in the seasonal leaf age distribution between the 194 trees monitored
by the phenocam and the 36 trees monitored at the branch level (Fig. 4) is in agreement
with the results presented by Gonçalves et al. (2020), which showed that the leaf phenology
and demography of the 194 trees are representative of the region of this study, by
comparing it to corresponding satellite vegetation indices retrieved from MODIS-MAIAC
(Multi-Angle Implementation of Atmospheric Correction). Also, this, together with the fact
that the isoprene trait distribution was similar among the scales (leaf level and upper
canopy), implies that the $E_s$ variation with leaf age measured here can be used to optimize
model estimates for intra- and inter-annual isoprene emission.
*3.3 Modeling of isoprene emission*
We used MEGAN to estimate isoprene emissions for the periods that we have in-situ
observations of isoprene and model inputs without considerable gaps, i.e., the years 2014
and 2015. We performed four different simulations (Fig. 5 and Table 2). For our first
simulation (S1), we applied MEGAN default settings for tropical vegetation (Fig. 5c,d),
which means that we used the $E_s$ assigned to the broadleaf evergreen tropical tree and the
broadleaf deciduous tropical tree that is equal to 7 mg m$^{-2}$ h$^{-1}$ (Guenther et al., 2012), half-
hourly averages of air temperature and PPFD data measured at the same tower as the
isoprene observations (Fig. 5a,b), and no change in the leaf age algorithm. For the second
simulation (S2), we used a modified  leaf age algorithm by adding the monthly distribution
of the LAI fractionated into leaf age classes (young, growing, mature, and old) as described
in the section above (Fig. 5c,d).
For a direct comparison between observations and model simulations, we performed eddy
covariance (EC) isoprene flux measurements during 11 days during Nov 2015 and
compared them with the simulations (Fig. 6). The isoprene emission sensitivity to the PPFD
circadian cycle was well simulated by MEGAN when estimates were compared with EC
isoprene flux ($r^2$=0.84, p-value <0.01) (Fig. 6 g). However, MEGAN simulations (S1 and
S2) overestimated the magnitude of emissions when compared with EC isoprene flux
around noontime (Fig. 6b); S1 and S2 had a daily average flux 2.71 and 2.68 times higher
than EC isoprene flux (p<0.01), respectively (Fig. 6h). This overestimation was a result of
a high value for $E_s$ in the model setup (7 mg m$^{-2}$ h$^{-1}$). To support this finding, we calculated
$E_s$ from the observed EC isoprene flux data from 06:00 to 18:00 with the G93 algorithm
(Guenther et al., 1993), and $E_s$ resulted in 3.21±1.76 mg m$^{-2}$ h$^{-1}$. We then ran a third
simulation (S3) with the corrected $E_s$ (3.21 mg m$^{-2}$ h$^{-1}$) (Fig. 5c,d; Fig. 6b) and S3 estimates
presented a daily average flux 1.23 higher than EC isoprene flux (p=0.013) (Fig. 6b,h). The
mean $E_s$ calculated from EC isoprene flux is in the same range as the $E_s$ observed for the
leaf level emissions of 21 trees (Fig. 3 and table S1), indicating that $E_s$ from this study site
is lower than the one set in the model default.



Another modification in the model was done based on our leaf-level measurements. In
section 3.2, we present the $E_s$ variation across leaf ages and suggest that the seasonal
variation in canopy leaf age distribution results in an emergent property to canopy seasonal
variation in $E_s$. With the LAI fractionated into leaf age classes (phenocam data) along the
year and the ratios of $E_s$ (leaf level measurements) between mature and young leaves,
mature and old leaves, and mature and growing leaves, we modified the leaf age emission
activity factor of the leaf age algorithm in MEGAN. The modified leaf age emission
activity factor accounts for lower values of $E_s$ in young and old leaves compared to mature
and growing leaves (Table 2). In our fourth simulation (S4) (Fig. 5c,d; Fig. 6b), we added
the modification in the leaf age emission activity factor, which led to a daily average 1.15
higher than EC isoprene flux (p=0.03) (Fig. 6 h).
To evaluate the effectiveness of our modifications in the model on intra- and inter-annual
timescales, we compared the isoprene mixing ratios observed at 38m height in all
campaigns performed in 2014 and 2015 with the four simulations. As our observations,
except for Nov 2015, are mixing ratios, it is only possible to indirectly compare with
MEGAN using an atmospheric model. However, considering that isoprene emission is
primarily driven by changes in light, temperature, and leaf phenology (Alves et al., 2018)
and that the variability of these factors was included in the model, we can still test the
comparability of the changes in the magnitudes from our measurements and simulations
that resulted from intra- and inter-annual variations. In figure 7, we show linear regressions
between observations and simulations. All datasets were filtered to the period between 12-
15h, local time, to evaluate the time of the day with maximum emission and high mixing
in the surface layer and to reduce variability in photochemical isoprene loss rates. Figure
7a shows daily hourly averages (12-15h, local time) of observed mixing ratios and the four
simulations for isoprene from Feb and Mar 2014, Aug 2014, and Oct 2015, and, apart from
the slope, all simulations were similarly and significantly comparable to observations
($r^2$=0.41 and $r^2$=0.42, p<<0.01). As significant day-to-day isoprene variability was
observed - also over other Amazon regions, with isoprene concentrations of similar
magnitudes occurring during both wet and dry seasons, likely resulting from the longer wet
season lifetimes of isoprene (Wells et al., 2022) - we averaged our datasets for each month
that we have observations and simulations. Figure 7b shows the monthly averages (12-15h,
local time) of mixing ratios and emission estimates for isoprene. We observed that our
modifications in the model improved the estimates (from $r^2$=0.76 to $r^2$=0.83). However, the
differences were less significant (p=0.08) compared to the linear regression with daily
hourly averages (p<<0.01) (Fig. 7a). We expect that if more isoprene flux data, especially
from long-term measurements, were available for comparison with our simulations, we
could have more significant results.
In general, the modifications for the $E_s$ (S3 and S4) and the leaf age activity factor (S4)
improved the estimates because they account for biological factors that have intra- and
inter-annual variations in this study site (Gonçalves et al., 2020), which represent a major
source of uncertainty in MEGAN (Niinemets et al., 2010). In this light, the main
improvement presented here resulted from the $E_s$ correction since our observations showed
that $E_s$ was less than half of the value in the model default settings and that $E_s$ varies
significantly among leaf ages. This is important because $E_s$ is a crucial factor in determining





the magnitudes of emission of a specific site, which may vary substantially in Amazonia.
Although a long-term canopy flux measurement study in central Amazonia indicated that
$E_s$ does not vary seasonally and argued that intra-annual changes in isoprene emission
resulted only from micrometeorological and LAI variations (Langford et al., 2022), other
studies in central Amazonia have shown that emission varies substantially in a relatively
small spatial scale and on topographic gradients (Gu et al., 2017; Batista et al., 2019); and,
more recently, leaf-level measurements have shown that $E_s$ varies within tree species both
seasonally and spatially, in particular when these species occur in different forest types and
topography (Gomes Alves et al., 2022).
*3.4 Implications of intra- and inter-annual variabilities in isoprene emission for modeling*
Despite the high variability within seasons, our results showed significant changes between
seasons. We corroborate previous studies indicating that intra-annual variability in isoprene
emission results from changes in solar radiation, temperature, and leaf phenology (e.g.,
Yáñez-Serrano et al., 2015; Alves et al., 2016, 2018), but we suggest that there is seasonal
variation in the ecosystem $E_s$ resulting from changes in canopy leaf age distribution and
that this may contribute to the seasonality in the magnitude of actual emission rates. Even
though we only derived the ecosystem $E_s$ from canopy isoprene flux measured in Nov 2015
- an El-niño year, when we compared the ecosystem $E_s$ to the values from leaves measured
in Oct-Nov 2017 (normal year), we observed both were in the same range. It is important
to note that leaf-level $E_s$ from Oct-Nov 2017 showed significant differences among leaf
ages, with maximum values for mature leaves, and those values were similar to the
ecosystem $E_s$ measured in Nov 2015. Nonetheless, it is also worth noting that Oct and Nov
(dry season and dry-to-wet transition seasons) are months with a substantially higher
fraction of mature leaves in the canopy compared to those from the wet and wet-to-dry-
transition seasons, meaning that the $E_s$ from mature leaves likely predominates the
ecosystem $E_s$ in Oct-Nov. In this sense, we suggest that understanding how the $E_s$ changes
over seasons due to leaf age composition within LAI will considerably improve model
estimates of intra-annual variations in isoprene. However, more long-term measurements
of canopy isoprene flux are needed to test it.
Surprisingly, inter-annual variabilities were less pronounced than intra-annual variability
when comparing normal years with the 2015-El-niño year. Our air temperature
measurements showed a significant increase during the dry season of 2015-El-niño year
compared to normal years. On a larger scale, regional land surface temperature retrieved
by satellite showed an increase of up to + 4 ∘C from Oct to Dec 2015 in the Amazon basin
(Jiménez-Muñoz et al., 2016), and that was accompanied by a significant negative
maximum climatological water deficit in 43% of the whole Amazon rainforest (Aragão et
al., 2018). Such stresses were expected to provide a stimulus for isoprene emission, as it is
already largely known that isoprene emission can increase with increasing temperature and
that some studies have also shown that emissions increase after moderate drought (e.g.,
Werner et al., 2021). However, our results did not show a significant increase in isoprene
mixing ratios in Oct 2015 compared to the dry seasons of previous years. Understanding
mechanisms of intra- and inter-annual variations in canopy emissions of isoprene is
essential for predicting their influence on atmospheric chemical-physical processes. For



example, the contribution of isoprene as a sink for hydroxyl radical (OH) varied seasonally
(Nölscher et al., 2016); however, it did not vary significantly when a normal year and the
2015-El-niño year were compared in this study site (Pfannerstill et al., 2018), leading to
the conclusion that these forests contributed to the emission of other compounds to cope
with the stress during the 2015-El-niño year, resulting in an effect on the atmospheric
oxidative capacity (Pfannerstill et al., 2021).

Some models predicted that higher temperatures and extended drought periods resulting
from climate change might increase global isoprene emissions (Pegoraro et al., 2006).
However, more recently, a synthesis of studies performed in the Amazon suggested that,
as the increase in temperature comes along with biomass loss given deforestation and forest
degradation, a decrease in isoprene emission from Amazonia may be expected (Yáñez-
Serrano et al., 2020). Interestingly, although isoprene emission was not considerably higher
in the dry season of the 2015-El-niño year, previous studies observed higher monoterpene
emissions compared to other dry seasons (Yáñez-Serrano et al., 2018) and even higher
monoterpene emissions in drier and warmer days of the 2015-El-niño dry season
(Pfannerstill et al., 2018). In addition, another study conducted in central Amazonia
reported that the heat in 2015/16 led to a shift in plant emissions to more reactive
monoterpenes such as β-ocimene and that at high temperatures, monoterpene emissions
can be decoupled from photosynthesis (Jardine et al., 2017). Recently, leaf-level $E_s$
measurements in hyperdominant tree species in this study site showed that photosynthesis
and isoprene decreased while monoterpenes and sesquiterpenes proportionally increased
in the dry season, suggesting that plants might have emitted heavier compounds to cope
with the stress caused by high temperatures and potentially drought (Gomes Alves et al.,
2022). We suggest that anomalies in isoprene emission during extreme years are less
expected than anomalies in emissions of monoterpenes and sesquiterpenes since plants may
also emit compounds from their storage pools when there is a limited carbon supply to
produce isoprene, as might be the case of plants reducing photosynthesis under heat and
drought stresses.

**Summary and conclusions**

Understanding mechanisms of intra- and inter-annual variations in canopy emissions of
isoprene from Amazonia is essential for predicting their influence on atmospheric
chemical-physical processes, especially when considering the role of Amazonia in the
global BVOC emission budget. Earlier studies presented seasonal isoprene emissions and
related them to the seasonality of temperature, solar radiation, and leaf phenology.
Nevertheless, to the best of our knowledge, this is the first study showing the $E_s$ variation
across leaf ages for several Amazonian tree species and the first attempt to represent the
effect on seasonal isoprene flux with a model parameterization. Also, by comparing
observations of normal years to the extreme 2015-El-niño year, we were able to show that
isoprene emission does not substantially increase as a result of higher temperatures. We
suggest that the stress caused by elevated temperatures and droughts in extreme years might
reduce the isoprene temperature dependence, which is not currently well represented in
modeling.



Even though there are uncertainties related to measurements and model simulations, the
results presented here suggest that $E_s$ varied seasonally and that this is a key factor in
improving model predictions. Additionally, previous studies showed that a distinguished
high monoterpene emission accompanies a non-pronounced increase in isoprene emission
in extreme years during the dry season at this study site, which is interesting to investigate
further since monoterpenes have higher reactivity in the atmosphere. Therefore, more
detailed and long-term measurements of isoprene and other BVOCs are encouraged to
improve our understanding of the intra- and inter-annual variability of BVOC emissions in
Amazonia, especially measurements that also account for biological factors that might
contribute to more mechanistic surface emission modeling and subsequently lead to better
predictions of atmospheric chemical-physical processes.
**Data availability**
Datasets are available upon request on https://attodata.org.
**Authors' contributions**
Eliane Gomes Alves has designed this study and performed the leaf-level measurements,
the statistical analysis of observational datasets, and the MEGAN simulations. Raoni
Santana and Cleo Quaresma have contributed to the analysis of the datasets of canopy
isoprene mixing ratios and of micrometeorology. Santiago Botía has contributed to the
analysis of the phenocam dataset and performed the MEGAN simulations. Tyeen Taylor
contributed new measurements of isoprene emissions from tropical tree species and the
imputation modeling of isoprene trait to the tree species monitored by the phenocam. Ana
Maria Yáñez-Serrano and Jürgen Kesselmeier have provided the canopy isoprene mixing
ratios dataset. Pedro Ivo Lembo Silveira de Assis and Giordane Martins have contributed
with the leaf age monitoring at the branch level. Rodrigo de Souza and Sergio Duvoisin
Junior contributed to the collection of isoprene samples measured at leaf-level. Alex
Guenther and Dasa Gu have contributed with the chemical analysis of isoprene samples
measured at leaf-level and the MEGAN simulations. Anywhere Tsokankunku and Matthias
Sörgel contributed with the dataset of eddy covariance isoprene flux. Bruce Nelson and
Davieliton Pinto contributed to the collection and the analysis of the phenocam dataset.
Shujiro Komiya contributed to analyzing the micrometeorology dataset to run the MEGAN
simulations. Diogo Martins contributed to the surface LiDAR data collection and analysis.
Bettina Weber and Cybelli Barbosa contributed with the temperature dataset to run the
MEGAN simulations. Michelle Robin contributed new measurements of isoprene
emissions from tropical tree species. Kenneth Feeley, Alvaro Duque, Viviana Lemos,
Maria Contreras, Alvaro Idarraga, Norberto Lopez, Chad Husby, and Brett Jestrow
contributed expert guidance, specimen curation, field assistance, and botanical
identifications for isoprene measurements from trees in botanic gardens and private
collections. All authors contributed to the writing of the manuscript.
**Competing interests**
The authors declare that they have no conflict of interest



**Acknowledgements**
We thank the National Institute of Amazonian Research (INPA) and the Max Planck
Institute for Biogeochemistry (MPI-BGC) for their continuous support. We acknowledge
the support by the ATTO project (German Federal Ministry of Education and Research,
BMBF funds 01LB1001A; Brazilian Ministry of Science, Technology, Innovation and
Communication; FINEP/MCTIC contract 01.11.01248.00); UEA and FAPEAM,
LBA/INPA and SDS/CEUC/RDS-Uatumã. TCT was supported by grant #NSF-PRFB-
1711997, and #NSF-1754163. We also truly thank Marta Sá and Paulo Ricardo Teixeira
for their work on checking the quality of the micrometeorology dataset and the INPA's
Microteorology Lab for providing the dataset. We acknowledge the helpful support for
isoprene measurements in botanic gardens by Santiago Madriñan of the Jardín Botánico
"Guillermo Piñeres", Ana María Benavides and Juan David Fernandes of the Jardín
Botanico de Medellin, Carl Lewis and Chad Husby of the Fairchild Botanic Garden, and
Patrick Griffith, Joanna Tucker Lima, and Michelle Barros of the Montgomery Botanical
Garden. We would like to especially thank the field assistants and all the people involved
in the logistic support of the ATTO project, who were all imperative for the development
of this study. We also thank all the indigenous communities that have been bravely
protecting the forest, and the riverside communities that have always helped us to do our
science. Without the "mateiros" we could never accomplish our scientific goals.

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

A New Field Instrument for Leaf Volatiles Reveals an Unexpected Vertical Profile of
Isoprenoid Emission Capacities in a Tropical Forest, Frontiers in Forests and Global
Change, 4, 1–22, https://doi.org/10.3389/ffgc.2021.668228, 2021.
Varshney, C. K. and Singh, A. P.: Isoprene emission from Indian trees, J Geophys Res,
108, 4803, https://doi.org/10.1029/2003JD003866, 2003.
Vickers, D. and Mahrt, L.: Quality control and flux sampling problems for tower and
aircraft data, J Atmos Ocean Technol, 14, 512–526, https://doi.org/10.1175/1520-
0426(1997)014<0512:QCAFSP>2.0.CO;2, 1997.
Wei, D., Fuentes, J. D., Gerken, T., Chamecki, M., Trowbridge, A. M., Stoy, P. C., Katul,
G. G., Fisch, G., Acevedo, O., Manzi, A., von Randow, C., and dos Santos, R. M. N.:
Environmental and biological controls on seasonal patterns of isoprene above a rain forest
in    central    Amazonia,    Agric    For    Meteorol,    256–257,    391–406,
https://doi.org/10.1016/j.agrformet.2018.03.024, 2018.
Wells, K. C., Millet, D. B., Payne, V. H., Vigouroux, C., Aquino, C. A. B., Mazière, M.,
Gouw, J. A., Graus, M., Kurosu, T., Warneke, C., and Wisthaler, A.: Next-Generation
Isoprene Measurements From Space: Detecting Daily Variability at High Resolution,
Journal    of    Geophysical    Research:    Atmospheres,    127,
https://doi.org/10.1029/2021JD036181, 2022.
Werner, C., Meredith, L. K., Ladd, S. N., Ingrisch, J., Kübert, A., van Haren, J., Bahn, M.,
Bailey, K., Bamberger, I., Beyer, M., Blomdahl, D., Byron, J., Daber, E., Deleeuw, J.,
Dippold, M. A., Fudyma, J., Gil-Loaiza, J., Honeker, L. K., Hu, J., Huang, J., Klüpfel, T.,
Krechmer, J., Kreuzwieser, J., Kühnhammer, K., Lehmann, M. M., Meeran, K., Misztal,
P. K., Ng, W.-R., Pfannerstill, E., Pugliese, G., Purser, G., Roscioli, J., Shi, L., Tfaily, M.,
and Williams, J.: Ecosystem fluxes during drought and recovery in an experimental forest,
Science (1979), 374, 1514–1518, https://doi.org/10.1126/science.abj6789, 2021.
Wu, J., Albert, L. P., Lopes, A. P., Restrepo-Coupe, N., Hayek, M., Wiedemann, K. T.,
Guan, K., Stark, S. C., Christoffersen, B., Prohaska, N., Tavares, J. v., Marostica, S.,
Kobayashi, H., Ferreira, M. L., Campos, K. S., Silva, R. da, Brando, P. M., Dye, D. G.,
Huxman, T. E., Huete, A. R., Nelson, B. W., and Saleska, S. R.: Leaf development and
demography explain photosynthetic seasonality in Amazon evergreen forests, Science
(1979), 351, 972–976, https://doi.org/10.1126/science.aad5068, 2016a.
Wu, J., Albert, L. P., Lopes, A. P., Restrepo-Coupe, N., Hayek, M., Wiedemann, K. T.,
Guan, K., Stark, S. C., Christoffersen, B., Prohaska, N., Tavares, J. v., Marostica, S.,
Kobayashi, H., Ferreira, M. L., Campos, K. S., da Silva, R., Brando, P. M., Dye, D. G.,



Huxman, T. E., Huete, A. R., Nelson, B. W., and Saleska, S. R.: Leaf development and
demography explain photosynthetic seasonality in Amazon evergreen forests, Science
(1979), 351, 972–976, https://doi.org/10.1126/science.aad5068, 2016b.
Yáñez-Serrano, A. M., Nölscher, A. C., Williams, J., Wolff, S., Alves, E., Martins, G. A.,
Bourtsoukidis, E., Brito, J., Jardine, K., Artaxo, P., and Kesselmeier, J.: Diel and seasonal
changes of biogenic volatile organic compounds within and above an Amazonian
rainforest, Atmos Chem Phys, 15, 3359–3378, https://doi.org/10.5194/acp-15-3359-2015,
1154  2015.
Yáñez-Serrano, A. M., Nölscher, A. C., Bourtsoukidis, E., Gomes Alves, E., Ganzeveld,
L., Bonn, B., Wolff, S., Sa, M., Yamasoe, M., Williams, J., Andreae, M. O., and
Kesselmeier, J.: Monoterpene chemical speciation in the Amazon tropical rainforest:
variation with season, height, and time of day at the Amazon Tall Tower Observatory
(ATTO), Atmos Chem Phys, 18, 3403–3418, https://doi.org/10.5194/acp-2017-817, 2018.
Yáñez-Serrano, A. M., Bourtsoukidis, E., Alves, E. G., Bauwens, M., Stavrakou, T., Llusià,
J., Filella, I., Guenther, A., Williams, J., Artaxo, P., Sindelarova, K., Doubalova, J.,
Kesselmeier, J., and Peñuelas, J.: Amazonian biogenic volatile organic compounds under
global change, Glob Chang Biol, 26, 4722–4751, https://doi.org/10.1111/gcb.15185, 2020.
Zannoni, N., Leppla, D., Lembo Silveira de Assis, P. I., Hoffmann, T., Sá, M., Araújo, A.,
and Williams, J.: Surprising chiral composition changes over the Amazon rainforest with
height, time and season, Commun Earth Environ, 1, 1–11, https://doi.org/10.1038/s43247-
1167  020-0007-9, 2020.




**Tables**

**Table 1.** Isoprene mixing ratios (ppbv) at 38 m for all field campaigns. Mixing ratios are mean values of isoprene measured at 12:00-15:00, local time (UTC-4h). Values within brackets are one standard deviation of the mean and the number of sampling days for each campaign.

| Year | Month | Season | Isoprene (ppbv) at 38 m |
|------|-------|--------|-------------------------|
| 2012 | November | dry-to-wet transition season | 9.30 (4.90) (n=4 days) |
| 2013 | February | wet season | 1.10 (0.66) (n=6 days) |
| 2013 | March | wet season | 1.84 (1.44) (n=3 days) |
| 2013 | June | wet-to-dry transition season | 1.83 (0.82) (n=5 days) |
| 2013 | September | dry season | 5.02 (1.99) (n=8 days) |
| 2014 | February | wet season | 5.92 (4.89) (n=3 days) |
| 2014 | March | wet season | 2.92 (2.50) (n=11 days) |
| 2014 | August | dry season | 7.76 (2.49) (n=15 days) |
| 2015 | October | dry season – *El-Niño* year | 8.94 (1.41) (n=13 days) |



**Table 2**. Model parameters for all simulations for the years 2014 and 2015.

| | 1st model simulation (S1) | 2nd model simulation (S2) | 3rd model simulation (S3) | 4th model simulation (S4) |
|---|---|---|---|---|
| PPFD and air temperature | 30 min averages – tower measurements | 30 min averages – tower measurements | 30 min averages – tower measurements | 30 min averages – tower measurements |
| $\beta^1$ | 0.13 | 0.13 | 0.13 | 0.13 |
| $LDF^2$ | 1 | 1 | 1 | 1 |
| $C_{t1}{}^3$ | 95 | 95 | 95 | 95 |
| $C_{eo}{}^4$ | 2 | 2 | 2 | 2 |
| Isoprene emission factor ($E_s$) | 7 mg m$^{-2}$ h$^{-1}$ | 7 mg m$^{-2}$ h$^{-1}$ | 3.21 mg m$^{-2}$ h$^{-1}$ | 3.21 mg m$^{-2}$ h$^{-1}$ |
| LAI | 5.32 | 5.32 | 5.32 | 5.32 |
| Leaf age algorithm – LAI | default | Modified with leaf age classes derived from the phenocam: *young leaves (0−1 month), growing (1−2 months), mature leaves (3−6 months), old leaves (>6 months).* | Modified with leaf age classes derived from the phenocam: *young leaves (0−1 month), growing (1−2 months), mature leaves (3−6 months), old leaves (>6 months).* | Modified with leaf age classes derived from the phenocam: *young leaves (0−1 month), growing (1−2 months), mature leaves (3−6 months), old leaves (>6 months).* |
| Leaf age emission activity factor | default $A_{new}$=0.05 $A_{gro}$=0.6 $A_{mat}$=1 $A_{old}$=0.9 | default $A_{new}$=0.05 $A_{gro}$=0.6 $A_{mat}$=1 $A_{old}$=0.9 | default $A_{new}$=0.05 $A_{gro}$=0.6 $A_{mat}$=1 $A_{old}$=0.9 | modified according to leaf-level measurements: $A_{new}$=0.01 $A_{gro}$=1 $A_{mat}$=1 $A_{old}$=0.64 |

*Note:* Empirical coefficients are from Guenther et al. (2012)
1. Temperature empirical coefficient
2. Light-dependent fraction
3. Temperature empirical coefficient
4. Emission-class dependent empirical coefficient





**Figures**

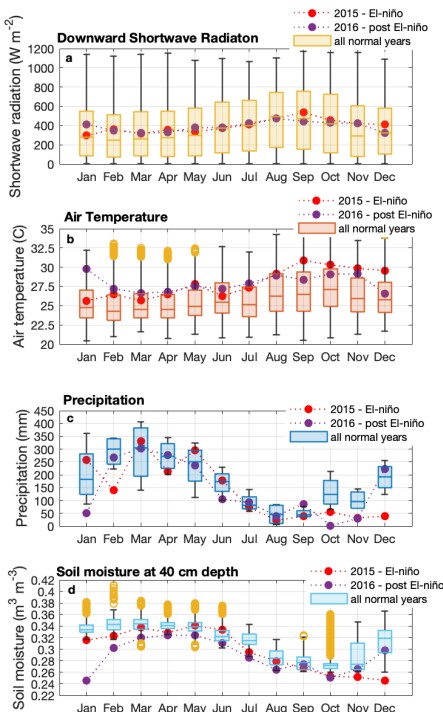

**Figure 1**. Seasonal variation of solar radiation (a), air temperature (b), precipitation (c), and soil
moisture (d) during normal years (2013, 2014, 2017, 2018, and 2019), an El-niño (2015), and post-
El-niño year (2016) - measured at the ATTO site. Boxplots present the median, the lower, and the
upper quartiles, where the upper quartile corresponds to the 0.75 quantile and the lower quartile
corresponds to the 0.25 quantile; whiskers connect the upper quartile and lower quartile to the
maximum and minimum nonoutliers, respectively; and outliers are values that are more than
1.5*IQR (interquartile range) away from the top or bottom of the box.

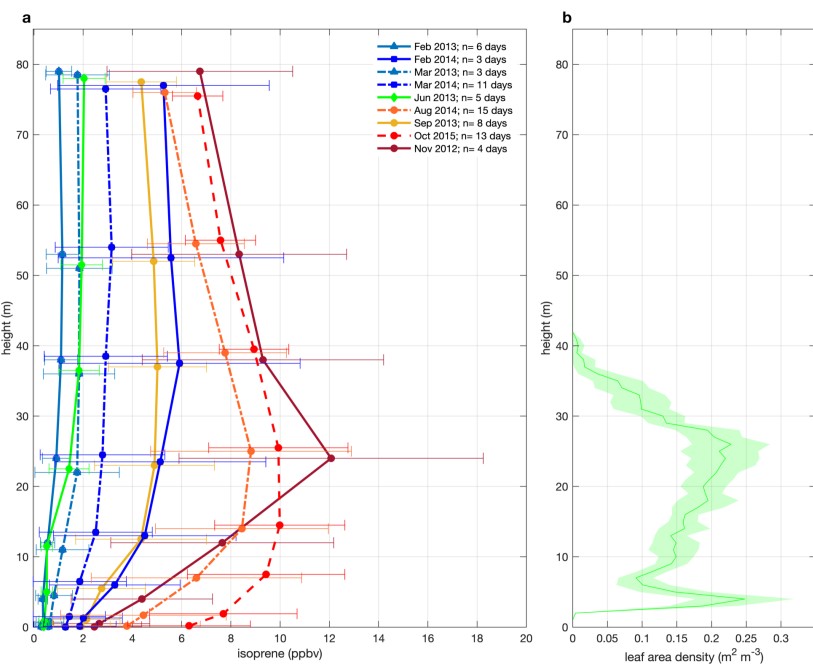

**Figure 2**. Mean isoprene mixing ratios for all field campaigns from Nov 2012 to Oct 2015, with
one standard deviation - 12:00-15:00 local time, UTC-4h - a daytime period that isoprene emission
is the highest; and mean canopy leaf area density profile with a confidence interval of 95% (b).
The measurements of all intensive campaigns were collected at the same heights (0.05, 0.5,
4, 12, 24, 38, 53, and 79 m), but note that in the plot (a) the heights were shifted by 50 cm
only for the better visualization of the error bars.




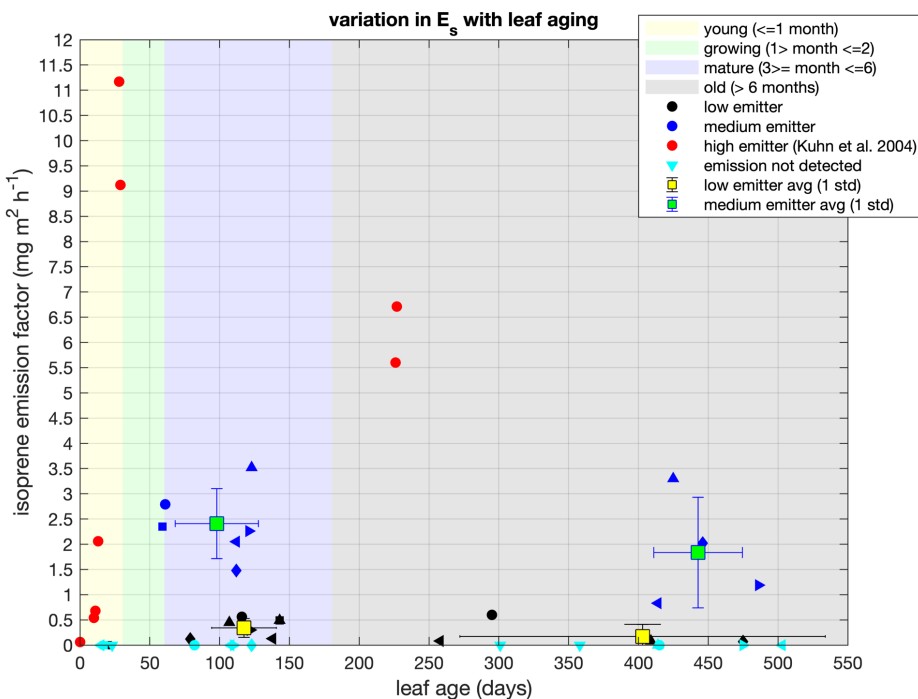

**Figure 3.** Isoprene emission factor ($E_s$) across leaf age classes and characterized into qualitative
emission categories – low, medium, and high.   Measured tree species were categorized into
medium (blue) and low (black) emitters according to their $E_s$ values, and different symbols
represent different tree species.   The high emitter category (red) represents a tropical species
measured in Kuhn et al. (2004b). Values represent observations of individual trees, and mean and
one standard deviation for the categories medium and low emitters at mature and old leaf age
classes. Shade areas represent the intervals of days for each leaf age class.





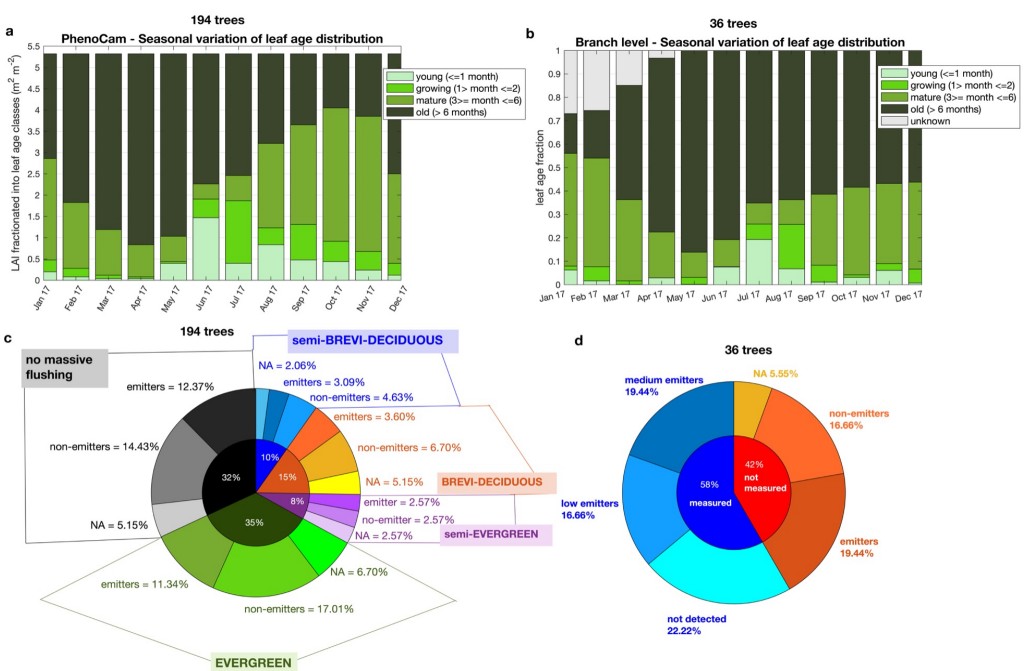


**Figure 4.** Leaf phenology and demography and isoprene emission trait. Panel (a) shows the leaf age distribution fractionated into LAI that was observed with the phenocam, in 2017; and panel (b) shows the leaf age distribution observed at branch level for 36 trees, in 2017 - note that unknown age refers to leaves that were attached to the branch at the beginning of monitoring and therefore could not be assigned to an age class. Panel (c) shows the percentual distribution of the phenotypes assigned to the 194 trees observed with the phenocam – no massive flushing, evergreen, semi-evergreen, deciduous, and semi-brevideciduous –, and the emission trait assigned to each tree species within these phenotypes – emitters, non-emitters, and NA (NA=no data available). Panel (d) presents the percentual distribution of the isoprene trait estimated to the non-measured trees (red); and the isoprene emission trait within measured tree species (blue), with measured tree species being categorized in classes of medium emission, low emission and not detected emission.



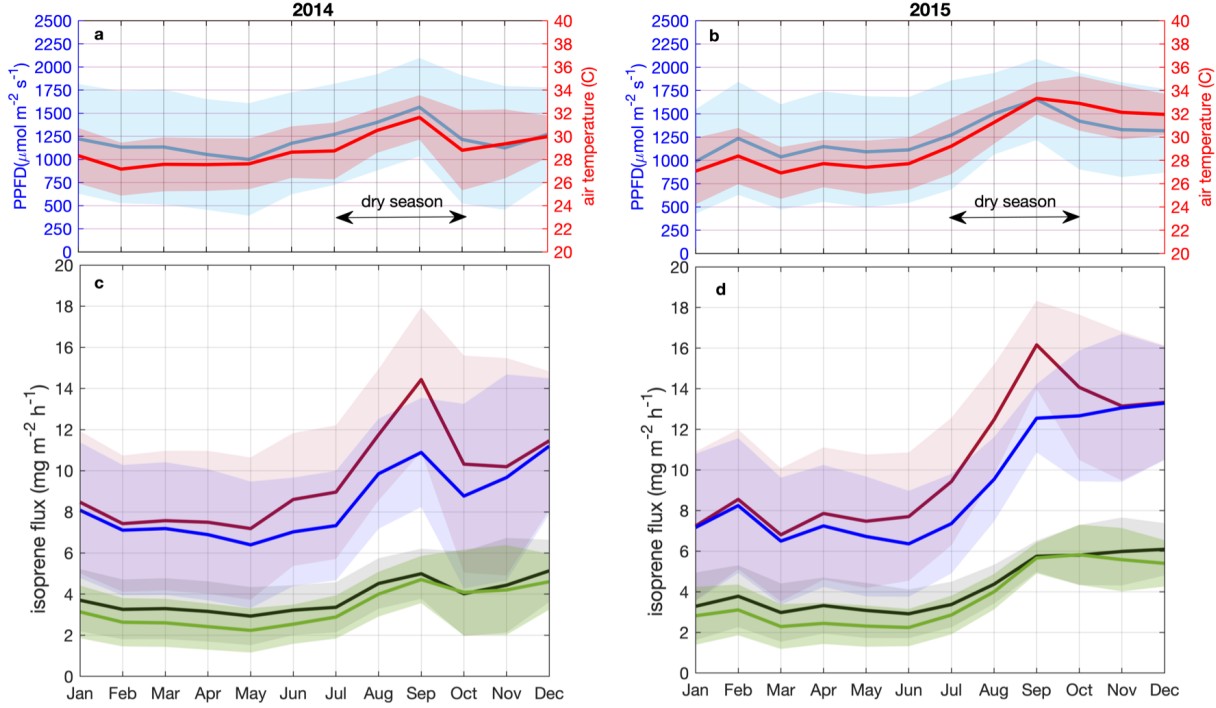

S1: MEGAN simulation default (no change in the LAI code, EF=7 mg m$^{-2}$ h$^{-1}$ and leaf age activity factor unmodified)

S2: MEGAN simulation with change for LAI - leaf age classes, EF=7 mg m$^{-2}$ h$^{-1}$ and leaf age activity factor unmodified

S3: MEGAN simulation with change for LAI - leaf age classes, EF=3.21 mg m$^{-2}$ h$^{-1}$ and leaf age activity factor unmodified

S4: MEGAN simulation with change for LAI - leaf age classes, EF=3.21 mg m$^{-2}$ h$^{-1}$ and leaf age activity factor modified

**Figure 5**. Simulated isoprene emission flux for 2014 and 2015. Monthly average of PPFD and air temperature (a, b) measured at the INSTANT tower. Simulations for 2014 (c) and 2015 (d) are: MEGAN simulation default, no change in the LAI code, emission factor equals to 7 mg m$^{-2}$ h$^{-1}$ and leaf age activity factor unmodified - S1; MEGAN simulation with change for LAI - leaf age classes, emission factor equals to 7 mg m$^{-2}$ h$^{-1}$ and leaf age activity factor unmodified - S2; MEGAN simulation with change for LAI - leaf age classes, emission factor equals to 3.21 mg m$^{-2}$ h$^{-1}$ and leaf age activity factor unmodified - S3; MEGAN simulation with change for LAI - leaf age classes, emission factor equals to 3.21 mg m$^{-2}$ h$^{-1}$ and leaf age activity factor modified - S4. Solid lines are means, and shaded areas represent one standard deviation of the mean.






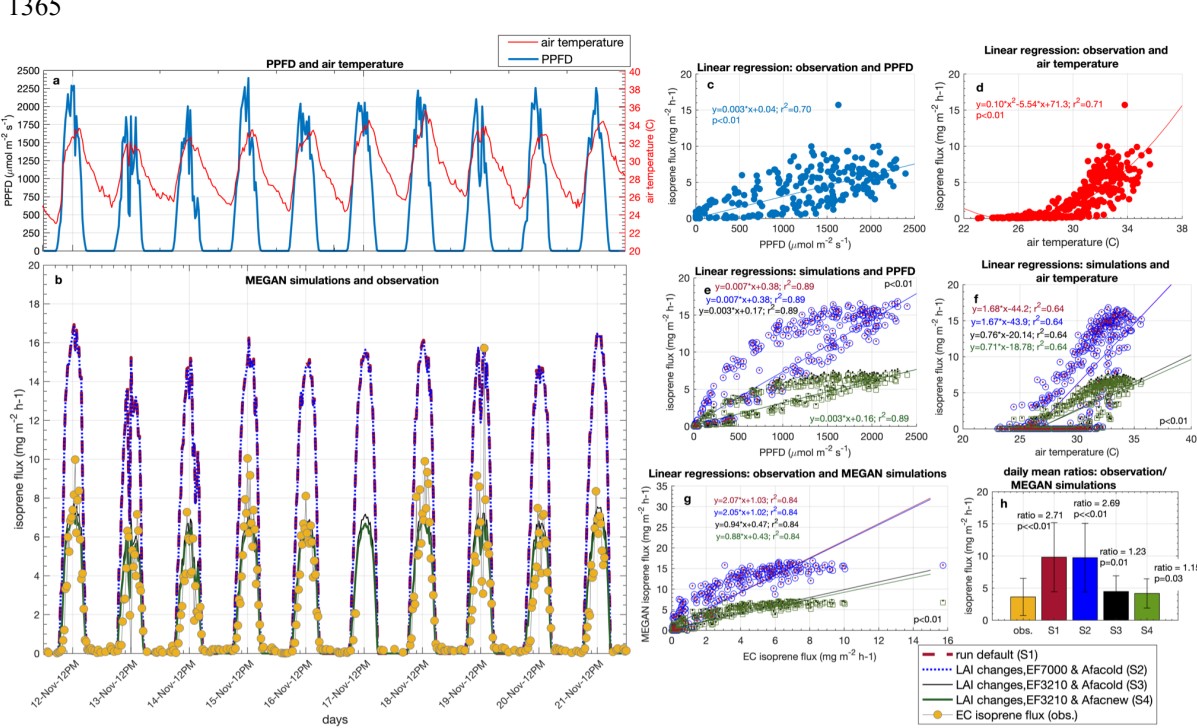

**Figure 6.** Observation of isoprene flux (eddy covariance) and MEGAN simulation for 11 days
in November 2015. Half-hourly averages of PPFD and air temperature (a); EC isoprene flux and
MEGAN simulations (b); linear regression between EC isoprene flux and PPFD (c); linear
regression between EC isoprene flux and air temperature (d); linear regression between simulations
and PPFD (e); linear regression between simulations and air temperature (f); linear regression
between EC isoprene flux and simulations (g); daily mean ratios between observation and
simulations (h).






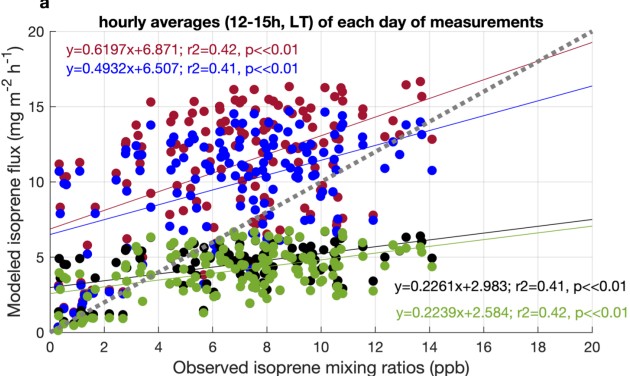

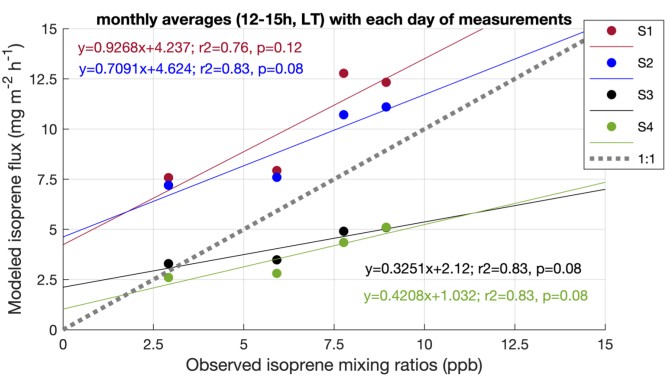

**Figure 7**. Correlation between isoprene mixing ratios observed at 38m during Feb and Mar
2014, Aug 2014, and Oct 2015, and the four simulations done for the respective periods.
Hourly averages (12-15h, local time (LT)) of each day of measurements (a); and monthly
averages (12-15h, local time (LT)) with each day of measurements (b).