# Peer review of "Intra- and inter-annual changes in isoprene emission from central Amazonia"

_EGUsphere, 2023_

## Author Comment (AC1)

**RC1:**
Well-written paper showing isoprene concentration and flux (11 days) measurements with PTR-TOF-MS and its comparison to various MEGAN model runs at ATTO site, central Amazonia. Paper shows for the first time emission factors of isoprene which differs by tree species and age. That makes this contribution special. In addition, authors did multiple MEGAN model runs to show comparison of measured fluxes of isoprene by eddy covariance and MEGAN with and without those specific Es. Results show substantial improvement; however, an overestimation of MEGAN model still remains. Authors discuss that the reason, among other, might be an emission of "heavier" compounds instead of isoprene.

Paper reads well and goes directly to the point. Maybe in discussion there should be mention the uncertainty of isoprene emissions by eddy covariance itself. If isoprene would be measured with H3O+ primary ions (which reader unfortunately does not know), then 232MBO (2-methyl-3-buten-2-ol) goes to m/z 69.07 too. However, maybe authors did their measurement in NO+ mode. Or isoprene might be fragmented and therefore you measured lower fluxes. Anyway, it should be clarified in M&M part and discussed properly.

**Reply:** We thank the reviewer for the time dedicated to reviewing this manuscript and for all the comments and suggestions. We acknowledge that some critical information is missing in M&M, and now we add them here and in the manuscript.

Our measurements were done with a quadrupole PTR-MS (not a PTR-TOF-MS) and H3O+ primary ions.

About the contribution of 232MBO (2-methyl-3-buten-2-ol) to m/z 69.07, we acknowledge that this is a known problem for the quadrupole systems. However, we have some critical aspects to consider in this study. MBO emissions are typically associated with conifer emissions (Kaser et al., 2013; Karl et al., 2014; Harley et al., 1998), which is not the typical vegetation of our study site. In tropical ecosystems and for isoprene-emitting species, MBO may be emitted at a rate that is ca. 1% of the isoprene emission rate (Guenther, 2013). Similarly, very low MBO to isoprene was reported for African tropical ecosystems (Liu et al., 2021), and, over the Amazon, model emissions of MBO are typically three orders of magnitude less than isoprene emissions (Barkley et al., 2008). Therefore, MBO has indeed interferences with the version of the PTRMS used in this study, but it is expected not to be very significant. This potential MBO interference could slightly overestimate the presented isoprene fluxes and mixing ratios, which means that actual isoprene emissions could be lower than what our measurements are indicating, and that supports our argument that isoprene emissions are probably lower than anticipated for this Amazonian site.

About the fragmentation of isoprene, there is indeed fragmentation, and we added more information in the M&M on the operation conditions that define the fragmentation pattern (2 mbar drift pressure, 600 V drift voltage, 127 Td). However, we were calibrating the PTRMS very frequently with the isoprene standard, and as we observed that the fragmentation did not vary significantly, we are confident about the derived mixing ratios. In addition, we made zero measurements every hour and did a cross-validation of the PTRMS measurements with tube samples analyzed in a GCMS system, which showed comparable results (Yáñez-Serrano et al., 2015).

Regarding the uncertainties of eddy covariance measurements, we calculated systematic and random uncertainties for the isoprene (m69) fluxes. Systematic errors lead to the failure to capture all of the largest transporting scales, typically leading to an underestimation of the flux. The random error is due to inadequate sampling of the main transporting eddies because of using too short a record length (Vickers et al., 2009). We used the method described by Finkelstein and Sims (2001) to estimate the random flux uncertainty. The integral turbulence scale (ITS) was defined as the cross-correlation first crossing 1/e with a

maximum correlation period of 10 s. Mean daytime uncertainties of eddy covariance isoprene flux were at most 15%.

[Figure]

**Figure 1.** Mean diel variation of isoprene flux and the flux error during the period of investigation.

[Figure]

**Figure 2.** Median diel variation of isoprene flux and the flux error during the period of investigation.

The main sources of systematic uncertainties were sonic anemometer tilt, spatial separation between the sonic anemometer and the inlet tube, lag time between the vertical wind speed component w and the isoprene concentration due to transport time of the air sample through a long tube from the tower to the PTRMS on the ground. We corrected for these sources of uncertainty by applying the standard methods of correction (coordinate rotation, spectral correction in the high and low frequency range, and time lag correction respectively) using the eddy covariance software package Eddypro (LI-COR Biosciences, Lincoln, USA). More details about these procedures and the data quality control can be found in Pfannerstill et al. (2018) and Pfannerstill et al. (2022).

**RC1:** Here are more detailed remarks:

**line 145:** 2380 mm of precipitation is average from 2013-2019?
**Reply:** No, this is from 1998-2019 and corresponds to the climatological average obtained from the TRMM dataset. As measurements from the ATTO site only started in 2012, there was not enough local data to provide the climatological precipitation average for the site, and this time series (2012– 2019) is highly affected by the 2015 El-Niño drought (Botía et al., 2022). Therefore, here we have used the TRMM climatological average (1998-2019) calculated by Botía et al. (2022).

**line 162:** please indicate the exact version of PTRMS by Ionicon, primary ions used, E/N ratio, Td. Please indicate calibration here too. Was isoprene present in the cylinder? Although it is in Yãnez-Serrano et al. (2015), it should be mentioned here too. In addition, version of the instrument and primary ions used is not mentioned even in Yãnez-Serrano et al. (2015).
**Reply:** We rewrote session 2.2, including the details requested by the reviewer, as follows: "Isoprene gradient mixing ratios were inferred by air samples collected from the INSTANT tower (80 m height, coordinates: S 02°08.7520' W 58°59.9920') at eight heights in and above the canopy (0.05, 0.5, 4, 12, 24, 38, 53 and 79 m) during intensive campaigns across different seasons from November 2012 to October 2015. Eight heated (50 C) and insulated inlets (fluorinated ethylene propylene - FEP, OD ⅜ in.) were connected to a quadrupole Proton Transfer Reaction – Mass Spectrometer (PTRMS) (Ionicon Analytic GmbH, Austria) - using the primary ion H3O+ and operated under standard conditions (2.2 mbar drift pressure, 600V drift voltage, 127 Td), which was housed in an air-conditioned container 10 m from the INSTANT tower. The inlets were guided to a valve system, switching every 2 min between the different heights, completing a full profile in 16 min. While an inlet was not sampled, it was flushed by a bypass pump at a flow rate of 16 lpm. Humidity-dependent calibrations (using bubbled synthetic zero air to dilute the standard, regulated as close as possible to ambient humidity conditions) were performed using a gas cylinder containing isoprene (m/z 69). The dilution steps ranged from 22 to 0.8 ppb. To determine the background signal for isoprene, a catalytic converter (Supelco, Inc. with platinum pellets heated to >400 ◦C) was used to convert ambient VOC to $CO_2$ +$H_2O$. The background signal was measured once every hour and then interpolated over the time of the measurements. The detection limit (LOD) for isoprene varied between 0.09 (wet season) and 0.1 (dry season) ppb. The mean total uncertainty of isoprene mixing ratios was 9.9 %, within the PTRMS measurement uncertainty (~10%). For more details on the experimental setup, PTRMS conditions, and calibration, we refer the reader to Yãnez-Serrano et al. (2015)".

**line 175:** indicate frequency of anemometer
**Reply:** We rewrote the sentence, including the detail requested by the reviewer, as follows: "A CSAT3 sonic anemometer (Campbell Scientific Inc., Logan, U.S.A.) measured the three-dimensional wind speed at high frequency (1 Hz) and was placed at a distance of 0.5 m from the isoprene inlet".

**line 244:** "was determined by laboratory analysis" - sounds odd
**Reply:** We rewrote the beginning of this paragraph as follows:
"Isoprene content in the sorbent cartridges was determined in the laboratory at the University of California (Irvine, U.S.A.)".

**line 372:** "Combretaceae" should be in italics too
**Reply:** We thank the reviewer for the comment and corrected it to *Combretaceae*

**fig 2:** fig would benefit from making x and y axes descriptions bigger
**Reply:** We have redone the figure with larger font size for the x and y axes.

[Figure]

Figure 2. Mean isoprene mixing ratios for all field campaigns from Nov 2012 to Oct 2015, with one standard deviation - 12:00-15:00 local time, UTC-4h - a daytime period that isoprene emission is the highest; and mean canopy leaf area density profile with a confidence interval of 95% (b). The measurements of all intensive campaigns were collected at the same heights (0.05, 0.5, 4, 12, 24, 38, 53, and 79 m), but note that in the plot (a), the heights were shifted by 50 cm only for the better visualization of the error bars.

**line 448:** what does mean "higher gross primary productivity (GPP) fluxes"? I think it should be without "fluxes"
**Reply:** We thank the reviewer for the comment. This was a mistake, given the gross primary productivity (GPP) definition. We now removed "fluxes".

**Extra references cited to respond to RC1:**

Barkley, M. P., Palmer, P. I., Kuhn, U., Kesselmeier, J., Chance, K., Kurosu, T. P., Martin, R. V., Helmig, D., and Guenther, A.: Net ecosystem fluxes of isoprene over tropical South America inferred from Global Ozone Monitoring Experiment (GOME) observations of HCHO columns, J Geophys Res, 113, D20304, https://doi.org/10.1029/2008JD009863, 2008.

Guenther, A.: Biological and Chemical Diversity of Biogenic Volatile Organic Emissions into the Atmosphere, ISRN Atmospheric Sciences, 2013, 1–27, https://doi.org/10.1155/2013/786290, 2013.

Harley, P., Fridd-Stroud, V., Greenberg, J., Guenther, A., and Vasconcellos, P.: Emission of 2-methyl-3-buten-2-ol by pines: A potentially large natural source of reactive carbon to the atmosphere, Journal of Geophysical Research: Atmospheres, 103, 25479–25486, https://doi.org/10.1029/98JD00820, 1998.

Karl, T., Kaser, L., and Turnipseed, A.: Eddy covariance measurements of isoprene and 232-MBO based on NO+ time-of-flight mass spectrometry, Int J Mass Spectrom, 365–366, 15–19, https://doi.org/10.1016/j.ijms.2013.12.002, 2014.

Kaser, L., Karl, T., Schnitzhofer, R., Graus, M., Herdlinger-Blatt, I. S., DiGangi, J. P., Sive, B., Turnipseed, A., Hornbrook, R. S., Zheng, W., Flocke, F. M., Guenther, A., Keutsch, F. N., Apel, E., and Hansel, A.: Comparison of different real time VOC measurement techniques in a ponderosa pine forest, Atmos Chem Phys, 12, 27955–27988, https://doi.org/10.5194/acpd-12-27955-2012, 2013.

Liu, Y., Schallhart, S., Taipale, D., Tykkä, T., Räsänen, M., Merbold, L., Hellén, H., and Pellikka, P.: Seasonal and diurnal variations in biogenic volatile organic compounds in highland and lowland ecosystems in southern Kenya, Atmos Chem Phys, 21, 14761–14787, https://doi.org/10.5194/acp-21-14761-2021, 2021.

Finkelstein Peter L., Sims Pamela F. Sampling error in eddy correlation flux measurements. J. Geophys. Res.: Atmosphere. 2001;106(D4) Wiley-Blackwell: 3503–9.

Pfannerstill EY, Nölscher AC, Yáñez-Serrano AM, Bourtsoukidis E, Keßel S, Janssen RHH, Tsokankunku A, Wolff S, Sörgel M, Sá MO, Araújo A, Walter D, Lavrič J, Dias-Júnior CQ, Kesselmeier J and Williams J (2022) Corrigendum: Total OH Reactivity Changes Over the Amazon Rainforest During an El Niño Event. Front. For. Glob. Change 5:952123. doi: 10.3389/ffgc.2022.952123

Pfannerstill, E. Y., Nölscher, A. C., Yáñez-Serrano, A. M., Bourtsoukidis, E., Keßel, S., Janssen, R. H. H., Tsokankunku, A., Wolff, S., Sörgel, M., Sá, M. O., Araújo, A., Walter, D., Lavric, J., Dias-Júnior, C. Q., Kesselmeier, J., and Williams, J.: Total OH Reactivity Changes Over the Amazon Rainforest During an El Niño Event, Front. For. Glob. Change, 1, 600, https://doi.org/10.3389/ffgc.2018.00012, 2018.

Vickers, D., Thomas, C. and Law, B. 2009. Random and systematic CO2 flux sampling errors for tower measurements over forests in the convective boundary layer. Agric. Forest Meteor. 149, 73–83.

**RC2:** Overall, this paper presents some very interesting data about leaf age/phenology and its impacts on canopy isoprene emissions in an Amazonian rainforest. The seasonality of tropical isoprene emissions is complicated, with significant changes in the drivers of light and temperature. But there has been much speculation about other changes due to water status and leaf age. This study provides information about both whole-canopy phenology by camera and also leaf-level studies of both age and emission. The manuscript is overall well written and clear. The are only a moderate number of minor mistakes and the figures are visually effective.

In addition to some relatively minor points, my major concern is with the interpretation of the isoprene concentration data. Using canopy measurements of isoprene concentrations as a proxy for whole-canopy flux is always problematic and is particularly difficult given the micrometeorological complexity of a tropical rainforest canopy. Small differences in canopy height and ground topography can lead to tricky situations. Also, the high levels of total heat flux and the complicated dynamics around the Bowen ratio, (sensible/latent heat) make vertical mixing an important process for interpreting isoprene concentrations. My 'major corrections' comments are mainly focused on the simplistic interpretation of concentrations

as a proxy for flux. While I do not believe that addresses these comments will lead to substantial changes in the manuscript, I am requesting major revisions due to the underlying scientific concern.

**Reply:** We thank the reviewer for the time dedicated to reviewing this manuscript and for all the comments and suggestions. Below we address each point, including the concern regarding the interpretation of isoprene mixing ratios and canopy flux/micrometeorology.

**RC2:** Major corrections

**Lines 420-431:** need to discuss and consider the impact of micrometeorology on the isoprene concentration profiles. The concentration profiles are a combination of emissions and mixing. During a dry year, the Bowen ratio shifts. There is less moisture available throughout the canopy, and more sensible heat flux is generated versus latent heat flux. More sensible heat leads to more intense vertical mixing. You should consider this effect and there may be ancillary data to test if there was increased vertical mixing.

**Reply:** We acknowledge the impact of micrometeorology on the canopy profile of isoprene mixing ratios, and, following the reviewer's suggestion, we better discuss it with ancillary data.

We agree with the reviewer that the isoprene mixing ratio profiles are a combination of emissions and air mixing. We followed the suggestion of looking into the Bowen ratio across seasons. Unfortunately, ATTO only started having in-canopy heat and latent flux measurements in 2022. Therefore, we analyzed the 2022 datasets of all months we have field campaigns in the past (Feb, Mar, Aug, Sep, Oct, and Nov) and added these extra plots to Supplementary Information. Figure S2 shows the Bowen ratio (bottom panel), latent heat flux (middle panel), and sensible heat flux (top panel) measured at 24 m.

[Figure]

Figure S2: Diurnal cycle (07:00-17:00h, local time) of sensible (a) and latent heat (b) fluxes in Wm$^{-2}$ using data averaged every 30 mins processed with EddyPro Software, and the Bowen ratio calculated for this period (c). Colored lines represent their respective month in 2022, and the dry season is represented by a solid line with points, whereas the dashed lines with triangles refer to the wet season.

We observed that both sensible and latent heat fluxes increased in the dry season, resulting in a very similar Bowen ratio between seasons, especially for the daytime period in which we analyzed isoprene mixing ratios (12:00-15:00, local time). We also considered another parameter that infers the in-canopy stability. For that, we analyzed the canopy profiles of the potential temperature of all months that we had campaigns in 2013, 2014, and 2015 and normalized them to the temperature measured at the top canopy (36 m), accounting for the daytime period that we analyzed isoprene mixing ratios (12:00-15:00, local time) (Figure S3).

[Figure]

Figure S3: Mean vertical profiles of the potential temperature ($\theta$) in °C (12:00-15:00 local time), normalized to the potential temperature ($\theta_c$) measured at the mean canopy height (36 m). The heights represented are 4, 12, 26, 36, 40, 55, 73, and 81m.

We observed that for all seasons across the years, the atmosphere was stable below 24m, neutral between 24 and 40 m, and unstable above 40 m. These in-canopy stability profiles and Bowen ratio values suggest that, although air mixing might affect isoprene mixing ratio profiles, the higher isoprene concentrations in specific heights (e.g., 24 m in Nov 2012) result from higher emissions; similarly, the lower isoprene concentrations in the same specific heights, but different seasons, result from lower emissions.

We are aware that it is impossible to directly compare canopy mixing ratios and emissions. However, with these analyses of the Bowen ratio and the in-canopy stability, we are confident that seasonal changes in shape and magnitudes of the canopy profiles of isoprene mixing ratios are a consequence of seasonal changes in emission, meaning that we can indirectly infer changes in emissions with changes in isoprene mixing ratios.

We propose to add the above figures to the Supplementary Information of this manuscript, and we refer to these analyses in the Results and Discussion section. More details on how this has been changed in the text are presented in the following responses.

**Lines 438-440:** Again, need to consider changes in micrometeorology, not just sources.
**Reply:** We have rewritten this entire paragraph by adding the discussion on the analyses done with the Bowen ratio and in-canopy stability, as follows:
Lines 433-456: "Another interesting result was the seasonal variation in the shape of the isoprene mixing ratio profiles (Fig. 2a). In general, all wet seasons (Feb-Mar 2013/2014) and the wet-to-dry transition season (Jun 2013) data showed a constant profile with no clear vertical gradient of isoprene. On the other hand, the dry seasons (Sep 2013, Aug 2014, and Oct 2015) showed maximum mixing ratios between 12 m and 24m, and the dry-to-wet transition season (Nov 2012) presented a well-defined peak at 24 m. This variation in the shape of the isoprene mixing ratio profiles likely resulted from changes in isoprene emission across seasons. Even though isoprene mixing ratio profiles are a combination of emission and air mixing, when we analyzed the Bowen ratio at 24 m (figure S2) and the potential temperature profiles (4-81 m; figure S1) across seasons, we observed that in-canopy air mixing and the atmospheric stability were similar among seasons. This implies that changes in isoprene mixing ratio profiles were predominantly attributed to the increase in emission in certain layers, mostly at the upper canopy, during the dry and dry-to-wet transition seasons. Furthermore, we suggest that the process that results in variation in the shape of isoprene mixing ratio profiles is a combination of variations in the canopy leaf area density profile and canopy leaf age distribution throughout the year. The total amount of LAI has a small variation over the year; still, the fractions of leaf ages that compose this total LAI changes seasonally (Wu et al., 2016), as well as the shape of the canopy leaf area density profile, with significant changes at the upper canopy (Martins Rosa, 2016). During the wet-to-dry transition season (May-Jun) and the dry season (Jul- Oct), upper canopy trees presented leaf abscission and leaf flushing (Lopes et al., 2016, Gonçalves et al., 2020), and the maturing process on the following months toward the beginning of the wet season (Nov-Jan) might translate into higher leaf area density at the upper canopy (Martins Rosa, 2016) and higher gross primary productivity (GPP) fluxes (Botía et al., 2022). This implies that two processes might be simultaneously occurring: one is that when there are more leaves at the upper canopy, less light penetrates the canopy, which might induce the maximum isoprene emission at the upper canopy as observed in Nov 2012; the other one is that leaves at the upper canopy can have higher photosynthesis rates and, consequently, a higher isoprene emission factor when they are mature (Alves et al., 2014), and more mature leaves and higher GPP were observed in this study site during the dry-to-wet transition season and beginning of the wet season (Lopes et al., 2016; Gonçalves et al., 2020; Botía et al., 2022)".

**Lines 441-456:** this entire section is very speculative and post hoc. Need to be much more cautious and give some caveats. Overall, you are trying to interpret observations instead of testing hypotheses.
**Reply:** We disagree with the reviewer. Here we interpreted and discuss our results with other studies, but that were conducted in the same study site and with partially same dataset (e.g., leaf phenology). Previous studies in ATTO showed changes in leaf phenology, indicating changes in canopy leaf biomass (Lopes et al., 2016; Gonçalves et al., 2020), which were closely related to changes in GPP (Botía et al., 2022). Our study is adding another layer to these processes by showing that isoprene emission is also affected by these seasonal processes in leaf dynamics and GPP, and offers a more process-understanding of isoprene emissions in ATTO. This entire paragraph was rewriten with the addition of some micrometeoroly aspects and is shown in the above response.

**Lines 613-617:** again, you are ignoring the impact of vertical mixing. As discussed above, temperature, radiation and moisture availability all change vertical mixing and these factors also affect isoprene emissions. That means equating mixing ratios to fluxes is quite problematic.

**Reply:** We have rewritten that as follows:

Lines 609-617: "To evaluate the effectiveness of our modifications in the model on intra- and inter-annual timescales, we compared the isoprene mixing ratios observed at 38m height in all campaigns performed in 2014 and 2015 with the four simulations. As our observations, except for Nov 2015, are mixing ratios, it is only possible to indirectly compare with MEGAN using an atmospheric model. However, considering that: air mixing and atmospheric stability were similar among the seasons (figures S2 and S3); isoprene emission is primarily driven by changes in light, temperature, and leaf phenology (Alves et al., 2018); and the variability of these factors was included in the model; we can still test the comparability of the changes in the magnitudes from our measurements and simulations that resulted from intra- and inter-annual variations".

**Lines 685-686:** again, this could be due to enhanced vertical mixing during the dry period.

**Reply:** We have rewritten that as follows:

Lines 675-686: "Surprisingly, inter-annual variabilities were less pronounced than intra-annual variability when comparing normal years with the 2015-El-niño year. Our air temperature measurements showed a significant increase during the dry season of 2015-El-niño year compared to normal years. On a larger scale, regional land surface temperature retrieved by satellite showed an increase of up to + 4 ∘C from Oct to Dec 2015 in the Amazon basin (Jiménez-Muñoz et al., 2016), and that was accompanied by a significant negative maximum climatological water deficit in 43% of the whole Amazon rainforest (Aragão et al., 2018). Such stresses were expected to provide a stimulus for isoprene emission, as it is already largely known that isoprene emission can increase with increasing temperature and that some studies have also shown that emissions increase after moderate drought (e.g., Werner et al., 2021, Byron et al., 2022). However, our results did not show a significant increase in isoprene mixing ratios in Oct 2015 compared to the dry seasons of previous years, indicating that emissions were lower in Oct 2015, with the isoprene mixing ratio profiles unlikely affected by in-canopy air mixing changes as suggested by the in-canopy atmospheric stability analysis (figure S3) ".

**RC2:** Minor corrections

**Line 148:** select another term for "utmost".

**Reply:** We rewrote the sentence: "Air masses arriving at the site predominantly come from the east…"

**Lines 160-163:** give the line inside/outside diameter and the flow rates. And state if there was a bypass flow for when the lines weren't being sampled.

**Reply:** To combine the comments from both reviewers, we rewrote this entire session as follows:

"Isoprene gradient mixing ratios were inferred by air samples collected from the INSTANT tower (80 m height, coordinates: S 02°08.7520' W 58°59.9920') at eight heights in and above the canopy (0.05, 0.5, 4, 12, 24, 38, 53 and 79 m) during intensive campaigns across different seasons from November 2012 to October 2015. Eight heated (50 C) and insulated

inlets (fluorinated ethylene propylene - FEP, OD ⅜ in.) were connected to a quadrupole Proton Transfer Reaction – Mass Spectrometer (PTRMS) (Ionicon Analytic GmbH, Austria) - using the primary ion H3O+ and operated under standard conditions (2.2 mbar drift pressure, 600V drift voltage, 127 Td), which was housed in an air-conditioned container 10 m from the INSTANT tower. The inlets were guided to a  valve system, switching every 2 min between the different heights, completing a full profile in 16 min. While an inlet was not sampled, it was flushed by a bypass pump at a flow rate of 16 lpm. Humidity-dependent calibrations (using bubbled synthetic zero air to dilute the standard, regulated as close as possible to ambient humidity conditions) were performed using a gas cylinder containing isoprene (m/z 69). The dilution steps ranged from 22 to 0.8 ppb. To determine the background signal for isoprene, a catalytic converter (Supelco, Inc. with platinum pellets heated to >400 ∘C) was used to convert ambient VOC  to $CO_2$ +$H_2O$. The background signal was measured once every hour and then interpolated over the time of the measurements. The detection limit (LOD) for isoprene varied between 0.09 (wet season) and 0.1  (dry season)  ppb. The mean total uncertainty of isoprene mixing ratios was 9.9 %, within the PTRMS measurement uncertainty (~10%). For more details on the experimental setup, PTRMS conditions, and calibration, we refer the reader to Yãnez-Serrano et al. (2015)".

**Line 173:** 41 m is a relatively low sample height given a canopy height of 35 m. State why this inlet height was selected for the EC measurements.
**Reply:** During the campaign where we measured the isoprene reported here, we also measured several trace gases, both reactive and non-reactive, at 41m. In order to be able to compare and relate the measurements to each other, we placed the sampling inlets at the same location. Some of the measurements included reactive trace gases whose concentrations quickly go below the detection limit of the measuring instruments the higher above the canopy you go due to their reaction with above-canopy ozone (e.g. NO). Therefore, we picked 41m as a compromise to ensure that all measurements were above the detection limit of their instruments.

**Lines 197-199:** what was the spatial arrangement of the transects? Where they all parallel?
**Reply:** We added this information to the session as follows:
"Measurements of canopy leaf area density were carried out with a ground Light Detection and Ranging sensor (LiDAR) at the ATTO site. These measurements aimed to give information on the canopy structure around the INSTANT tower. Ground-LiDAR surveys were conducted in October 2015 with a Riegl LD90-3100VHS-FLP system (Horn, Austria), which generated a canopy profile map in vertical and horizontal directions. We walked ten transects of 150 m in length with the ground-LIDAR system. The transects were parallelly distributed at a distance of ~ 100 m from each other, with six transects to the east/northeast, three transects to the west, and one transect to the south of the INSTANT tower. Measurements were averaged every 15 m of each transect, summing up to ten measurements per transect. Measurements of all ten transects were then averaged and presented with the confidence interval (95%). More details about how the ground LiDAR data were analyzed can be obtained from Stark et al. (2012)".

**Lines 471**: Ontogeny refers to an organism. Phenology is the correct term for leaves, which are part of an organism and are cyclic.
**Reply:** we corrected the term to "phenological status"

**Lines 477-483:** you show the unscaled data in Figure 3 but describe the scaled data in the text. It would be nice to have a figure which shows the 36% reduction.
**Reply:** We have added a subplot with data in percentage to Figure 3.

[Figure]

Figure 3. Isoprene emission factor ($E_s$) across leaf age classes and characterized into qualitative emission categories – low, medium, and high. Measured tree species were categorized into medium (blue) and low (black) emitters according to their $E_s$ values, and different symbols represent different tree species. The high emitter category (red) represents a tropical species measured in Kuhn et al. (2004b). Values represent observations of individual trees and mean and one standard deviation for the categories medium and low emitters at mature and old leaf age classes. Shade areas represent the intervals of days for each leaf age class. The inset figure shows the mean $E_s$ ratios of mature (3-6 months) to young (0-1 month), growing (1-2 months), and old (> 6 months) leaves calculated from the ratio of each individual tree.

**Line 533:** "fractionated" select another term.
**Reply:** We have rewritten the sentence, as follows: "Here, with a dataset of 194 trees (Fig. 4, and table S2) monitored with a phenocam for leaf phenology and demography from 2013 to 2018, we derived: (i) the camera-based canopy leaf area index (LAI) **separated** into four leaf age classes…"

**Lines 588-590:** why use this procedure instead of simply finding adjust Es by 2.68 for S2? Then S3 would be 1 in line 592, correct?
**Reply:** We acknowledge that this paragraph needs a more detailed explanation. However, we disagree with the reviewer suggesting a simple correction of the $E_s$ by 2.68. This ignores

the fact that we have measurements from which the $E_s$ can be calculated, which is what we did for S3 using the G93 algorithm (Guenther et al. 1993). Furthermore, the $E_s$ is, by definition, the emission at standard conditions (1000 $\mu$mol m$^{-2}$ s$^{-1}$ photosynthetically active radiation- PAR, 30 ˚C) and a model input. So to derive an $E_s$, a more complex processing is required, rather than just using the overestimation factor of S2. Using a measurement-based $E_s$, we ran S3, which resulted in a better agreement with the observed values, and from this, we can confidently say that the previous $E_s$ in MEGAN was too high. The main purpose of this factorial simulation exercise is to highlight where the model can be improved based on measurements. A further point to note is that we got the $E_s$ from the G93 algorithm, and our EC measurements have a standard deviation of +/- 1.76 mg m$^{-2}$ h$^{-1}$. Within this variability, higher/lower $E_s$ cover the overestimation of 1.23 in S3 with respect to the EC flux data. And, to finalize, the relationship between daily average flux and $E_s$ is not linear.

**Lines 624-634:** Need to remove the monthly averaging: this is not a statistically sound approach. If you have a poor fit with your data, trying to average it away by temporal binning is incorrect. That's why your p value is not improving. And that's why only considering r2 isn't a good idea. You reduced to four points. If you had gone to 2, you'd have a perfect r2!
**Reply:** We removed the monthly averaging and have rewritten this paragraph, as follows:
Lines 617-634: "In figure 7, we show linear regressions between observations and simulations. All datasets were filtered to the period between 12-15h, local time, to evaluate the time of the day with maximum emission and high mixing in the surface layer and to reduce variability in photochemical isoprene loss rates. Figure 7 shows that, apart from the slope, all simulations were similarly and significantly comparable to observations (r2=0.41 and r2=0.42, p<<0.01). However, it is important to note that the finding of observed reduced $E_s$, compared to the model default settings, and its changing across leaf ages may have an important effect on isoprene intra-annual variation. Therefore,  we expect that if more isoprene flux data, especially from long-term measurements, were available for comparison with our simulations, we could have more significant results in comparing observations and the simulations with  all modifications in MEGAN (S4). Additionally, as significant day-to-day isoprene variability was observed - also over other Amazon regions, with isoprene concentrations of similar magnitudes occurring during both wet and dry seasons, likely resulting from the longer wet season lifetimes of isoprene (Wells et al., 2022), long-term flux measurements could help by offering a direct comparison between observations and modeling, and the possibility to evaluate atmospheric chemical processes".

**Lines 655-673:** work on this paragraph a bit. It has a strong beginning, but I am having trouble following the logic. The final two sentences are fine, but the logic in the middle is confusing and reasoning doesn't connect to the final statement.
**Reply:** We acknowledge that this paragraph had room for clarification, so we rewrote the paragraph as follows:
"Despite the high variability within seasons, our results showed significant changes between seasons. Previous studies have shown the strong seasonality of isoprene emission in central Amazonia, and we corroborate these studies that indicated changes in solar radiation, temperature, and leaf phenology, as the important drivers of isoprene intra-annual variability (e.g., Yáñez-Serrano et al., 2015; Alves et al., 2016, 2018). However, here we further develop our understanding concerning the effect of leaf phenology, by suggesting that there is seasonal variation in the ecosystem $E_s$ resulting from changes in canopy leaf age distribution, which may significantly contribute to the seasonality in the magnitude of actual emission rates. This is supported by our leaf-level $E_s$ measurements, which showed significant differences among leaf ages, with maximum values for mature leaves, and by our results on canopy leaf age distribution changes.  Furthermore, it is important to note that

leaf-level $E_s$ from Oct-Nov 2017 showed maximum values for mature leaves, and those were similar to the canopy $E_s$ measured in Nov 2015. Oct and Nov (dry season and dry-to-wet transition seasons) are months with a substantially higher fraction of mature leaves in the canopy compared to those from the wet and wet-to-dry transition seasons, meaning that the highest values of $E_s$ from mature leaves likely predominate the ecosystem $E_s$ in Oct-Nov. In this sense, understanding how the $E_s$ changes over seasons due to leaf age composition within LAI will considerably improve model estimates of intra-annual variations in isoprene. However, more long-term measurements of canopy isoprene flux are needed to test it".

**Lines 701-702:** is this from the current study? Do you mean isoprene concentrations?
**Reply:** Yes, it is from this study, and it refers to isoprene mixing ratios. The sentence has been rewritten as follows:
"Interestingly, although isoprene mixing ratios were not considerably higher in the dry season of the 2015-El-niño year, previous studies observed higher monoterpene mixing ratios compared to other dry seasons (Yáñez-Serrano et al., 2018) and even higher monoterpene mixing ratios in drier and warmer days of the 2015-El-niño dry season (Pfannerstill et al., 2018)".

**Line 1367 (Figure 6**): for panel (d), the title says linear regression, but the fit is quadratic. Also, for panel (h), it's the ratio of simulations to observations, not obs/MEGAN as stated.
**Reply:** We thank the reviewer for noticing that, we corrected the titles of panels (d) and (h) and the figure caption accordingly.

[Figure]

Figure 6. Observation of isoprene flux (eddy covariance) and MEGAN simulation for 11 days in November 2015. Half-hourly averages of PPFD and air temperature (a); EC isoprene flux and MEGAN simulations (b); linear regression between EC isoprene flux and PPFD (c); quadratic regression between EC isoprene flux and air temperature (d); linear regression between simulations and PPFD (e); linear regression between simulations and air temperature (f); linear regression between EC isoprene flux and simulations (g); daily mean ratios between simulations and observation (h).

**Line 1393 (Figure 7):** as mentioned above, remove panel (b)
**Reply:** Figure 7b was removed.

**Extra reference cited to respond to RC2:**
Byron, J.; Kreuzwieser, J.; Purser, G.; van Haren, J.; Ladd, S. N.; Meredith, L. K.; Werner, C.;
Williams, J.: Chiral monoterpenes reveal forest emission mechanisms and drought responses, Nature,
609, 7926, https://doi.org/10.1038/s41586-022-05020-5, 2022.